# RNA dysregulation in neurodegenerative diseases

Yini Li[1,2] & Shuying Sun [ID] [1,2,3 ✉]

## Abstract

**Dysregulation of RNA processing has in recent years emerged as a significant contributor to neurodegeneration. The diverse mechanisms and molecular functions underlying RNA processing underscore the essential role of RNA regulation in maintaining neuronal health and function. RNA molecules are bound by RNA-binding proteins (RBPs), and interactions between RNAs and RBPs are commonly affected in neurodegeneration. In this review, we highlight recent progress in understanding dysregulated RNA-processing pathways and the causes of RBP dysfunction across various neurodegenerative diseases. We discuss both established and emerging mechanisms of RNA-mediated neuropathogenesis in this rapidly evolving field. Furthermore, we explore the development of potential RNA-targeting therapeutic approaches for the treatment of neurodegenerative diseases.**

**Keywords** Neurodegenerative Diseases; RNA Metabolism; RNA-binding Protein; RNA-targeting Therapeutics
**Subject Categories** Neuroscience; Pharmacology & Drug Discovery; RNA Biology

See also: RNA review series

## Introduction

Neurodegenerative diseases are prevalent age-related diseases, with Alzheimer's disease (AD) as a prototype of such conditions. As of 2024, approximately 6.9 million Americans are affected by AD, making it the most common neurodegenerative disease, followed by Parkinson's disease (PD) (from the Alzheimer's Disease Association Report (2024)). There are also many less prevalent or rare neurodegenerative diseases such as Huntington's disease (HD), frontotemporal dementia (FTD) and amyotrophic lateral sclerosis (ALS). With the global demographic trend of population aging, it is predicted that an increasing number of people will be diagnosed with neurodegenerative diseases globally (Collaborators, 2024). Though the clinical symptoms of these diseases vary, multiple neurodegenerative diseases share similar underlying pathological mechanisms (Abramzon et al, 2020; Jellinger, 2010; Ling et al, 2013; Paulson, 2018; Wingo et al, 2022). The presence of pathological inclusions and causative mutations of RNA-binding proteins (RBPs) is increasingly observed among neurodegenerative diseases. In addition,

pathological repeat expansion in multiple diseases, such as ALS, FTD, HD and various types of spinocerebellar ataxia, yields repeat-containing RNAs that could cause neurotoxicity via various mechanisms (Paulson, 2018). In the post-genomic era, a variety of RNA processing pathways and emerging types of coding and noncoding RNAs have been commonly identified in the disease context (Statello et al, 2021), with potential contributions to neurodegeneration. Therapeutic strategies targeting RNA to modulate disease-linked genes have achieved significant success (Khorkova et al, 2023; Zhu et al, 2022).

Here, we focus on RNA-related pathogenic mechanisms in neurodegenerative diseases and updates on RNA-targeting therapeutic approaches that hold great promise. We review recent discoveries alongside previous key findings, aiming to offer a timely reference for research on RNA and neurodegenerative diseases, with a particular emphasis on ALS. We start with the various RNA processing pathways and provide representative examples of how these pathways are dysregulated in neurodegenerative diseases (Figs. 1–3). Next, we discuss the mechanisms that lead to RBP dysfunction (Fig. 4), resulting in dysregulation of RNA processing. Finally, we review the current progress in RNA-targeting therapeutics (Fig. 5). The different RNA processing pathways are often interconnected, and most RBPs have multifunctional roles across several RNA processing steps, creating significant interplay among them. Overall, these findings highlight RNA metabolism as a critical factor in disease mechanisms.

## RNA splicing

RNA splicing is a key regulatory step in gene expression during the RNA life cycle. In this process, introns are removed from the nascent pre-messenger RNA (pre-mRNA) and exons are joined to produce the fully processed mRNA. The vast majority of human genes (92–94%) undergo alternative splicing (Wang et al, 2008). Alternative splicing is disproportionally abundant and evolutionarily conserved in the brain (Barbosa-Morais et al, 2012), especially enriched in genes associated with highly specialized neuronal functions, such as synaptic transmission, axon guidance, actin cytoskeleton reorganization, and plasticity (Barbosa-Morais et al, 2012; Merkin et al, 2012; Wang et al, 2008).

Splicing dysregulation can be directly triggered by perturbed RBP function and altered RBP-spliceosome interaction due to disease-related mutations or pathologies. One example is TDP-43, related proteinopathies of which (i.e., nuclear clearance and cytosolic inclusion) are broadly found in about 97% of ALS, 50% of FTD, and 40-60% of AD patients (Ayala et al, 2008; Jo et al, 2020; Meneses et al, 2021). While mutations in TDP-43 account for 5% of familial ALS cases, TDP-43 pathology is also observed in

[1]Department of Physiology, Johns Hopkins University School of Medicine, Baltimore, MD 21205, USA. [2]Brain Science Institute, Johns Hopkins University School of Medicine, Baltimore, MD 21205, USA. [3]Departments of Neuroscience, Pathology, Johns Hopkins University School of Medicine, Baltimore, MD 21205, USA. ✉E-mail: shuying.sun@jhmi.edu

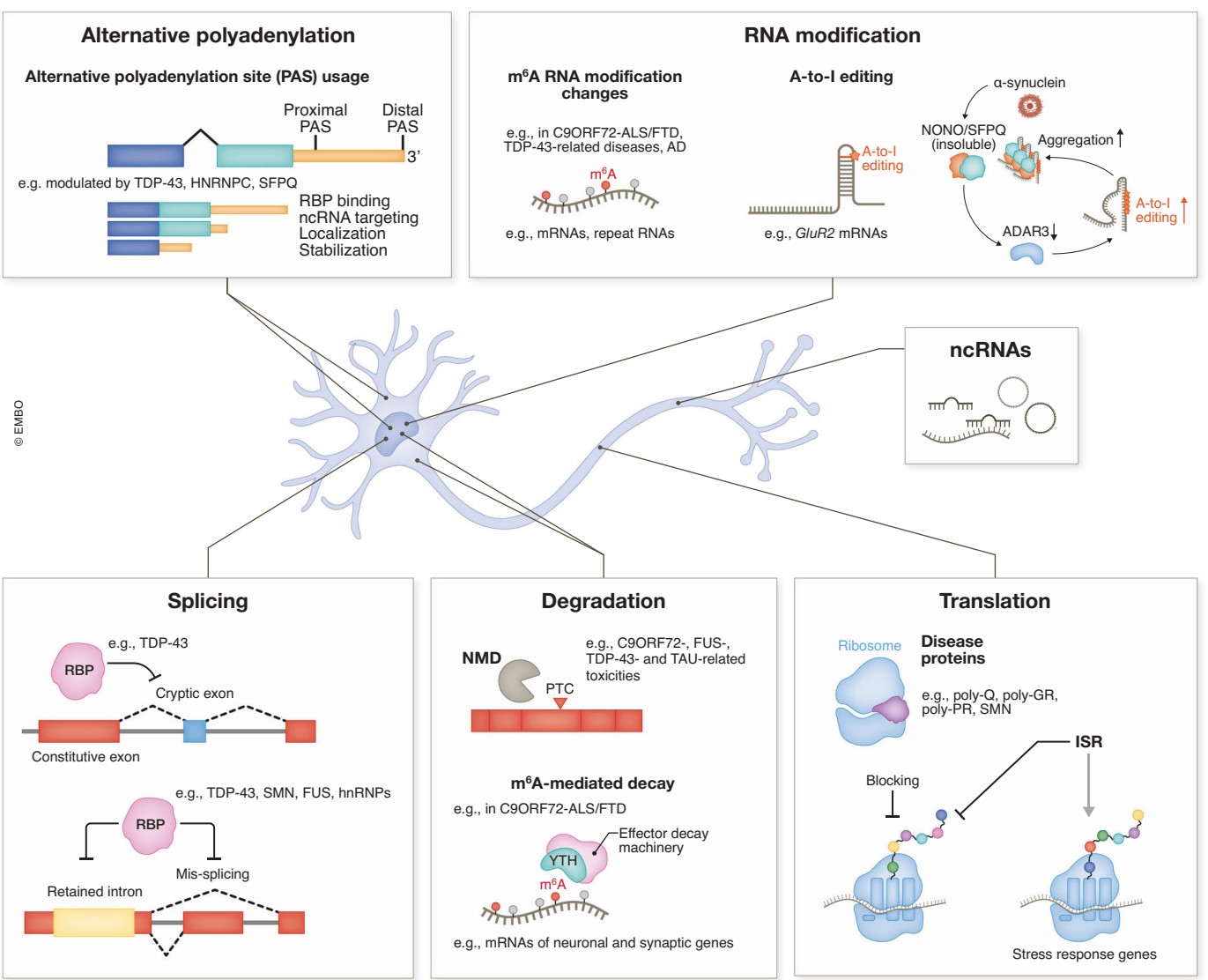

**Figure 1. Dysregulation of RNA processing in neurodegenerative diseases.**

RNA dysregulation can occur in almost all steps of RNA processing, including RNA modification, splicing, polyadenylation, degradation, and translation. The dysregulated RNAs can be protein-coding mRNAs or noncoding RNAs (ncRNAs).

sporadic disease. Increasing evidence suggests that the loss of nuclear TDP-43 is likely an early pathologic event preceding cytosolic aggregation (Sun et al, 2017; Vatsavayai et al, 2016). The function of TDP-43 on alternative splicing has been extensively investigated. Widespread splicing changes have been reported in TDP-43 loss-of-function in vitro and in vivo models (Polymenidou et al, 2011; Tollervey et al, 2011), and in models expressing causative TDP-43 mutations, such as TDP-43$^{Q331K}$ (Arnold et al, 2013) and TDP-43$^{M337V}$ (Watanabe et al, 2020).

Recently, TDP-43-regulated cryptic exons have widely attracted attention. Cryptic exons (CEs) represent a type of alternative splicing in which non-conserved intronic sequences are erroneously included in mature RNAs. Normally, binding of TDP-43 to intronic (UG)n-rich sequences suppresses the recognition of cryptic splice sites. With nuclear clearance of TDP-43, these splice sites are de-repressed, leading to the inclusion of cryptic exons in the mRNA (Fig. 2). Currently, about 100 CE-containing transcripts have been identified in TDP-43-deficient cells. Most identified TDP-43-mediated CEs cause frameshifts and introduce premature stop codons, while there are also some CEs linked to alternative transcription start sites, premature polyadenylation sites, and expansion of conserved exons of the mRNAs (Ling et al, 2015). Consequently, those transcripts containing splicing errors are often the targets of nonsense-mediated decay (NMD) or other surveillance RNA decay pathways. CEs in *STMN2 and UNC13A* are two examples that have been characterized in detail. Stathmin2 (*STMN2*) is a microtubule-related gene. The CE inclusion of *STMN2* exposes cryptic premature polyadenylation sites and reduces the levels of the functional full-length protein (Klim et al, 2019; Krus et al, 2022; Melamed et al, 2019; San Juan et al, 2022). Constitutive *Stmn2* knockout in mice results in delayed microtubule polymerization and axon outgrowth (Krus et al, 2022), and

## Splicing alteration in neurodegenerative diseases

### Dysfunction of the splicing machinery

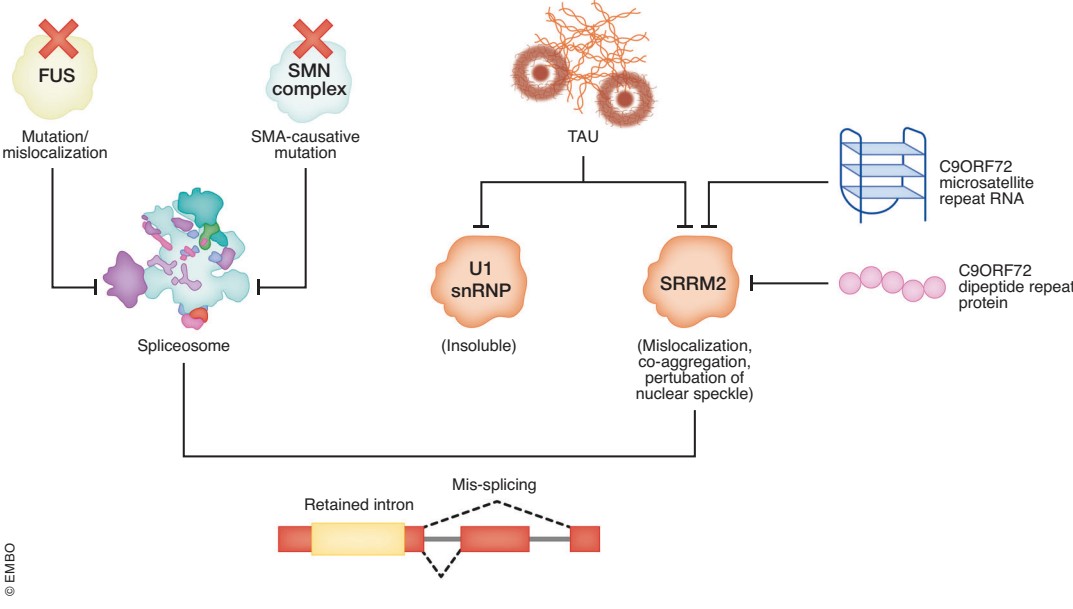

### Dysfunction of specific RBPs, such as TDP-43

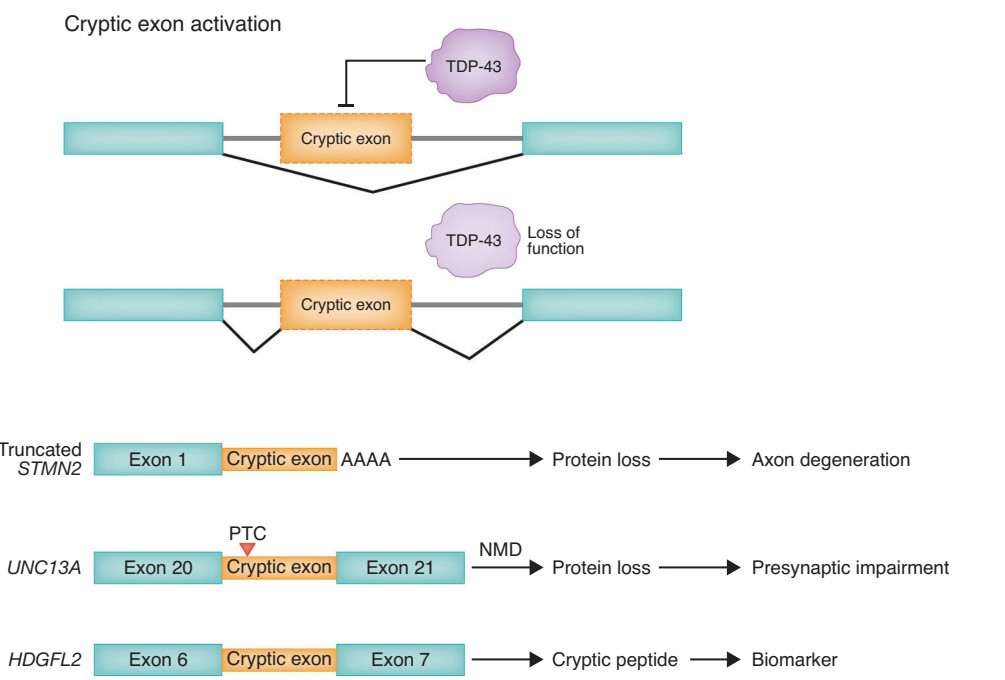

**Figure 2. Examples of splicing alterations in different disease contexts.**

Disease-related splicing changes can be caused by defects in the core splicing machinery, or by dysfunction of specific RNA-binding proteins, such as TDP-43.

persistent loss of stathmin-2 in adult mice results in ALS-linked pathologies, including reduced inter-neurofilament spacing, axonal caliber collapse, progressive motor and sensory deficits, and muscle denervation (Lopez-Erauskin et al, 2024; San Juan et al, 2022). Another example is *UNC13A*, a gene encoding a presynaptic protein. Loss of

*UNC13A* leads to synaptic defects in mice and uncoordinated movement behavior in *Caenorhabditis elegans* (Augustin et al, 1999; Sulston and Brenner, 1974). The out-of-frame CE inclusion results in transcripts degraded by NMD and decreases the functional protein level (Brown et al, 2022; Ma et al, 2022). In addition, TDP-43 pathology is also

commonly observed in AD (Meneses et al, 2021). As expected, a high frequency of CE inclusion was detected in a large cohort of AD brains (Hsieh et al, 2019; Sun et al, 2017). The CE of *STMN2* and *UNC13A* have been detected in TDP-43-associated AD (Agra Almeida Quadros et al, 2024). The study within the Religious Orders Study and Rush Memory and Aging Project (ROSMAP) also found that CE events concomitantly increase with Tau pathological burden (Hsieh et al, 2019).

On the other hand, isoforms with in-frame cryptic exons could lead to the production of proteins containing cryptic peptides in diseased neurons. These could be used as biomarkers for monitoring TDP-43 loss of function and disease stage. *HDGFL2* CE and its cryptic peptide is a good example. Elevated levels of HDGFL2 cryptic peptide can be detected in fluid samples from ALS/FTD patients, including those that carry disease-causative mutations but remain asymptomatic (Calliari et al, 2024; Irwin et al, 2024; Seddighi et al, 2024), showing great potential as a biomarker for ALS and FTD. Moreover, an increase in HDGFL2 levels appears to follow a pattern similar to that of the established biomarker neurofilament light chain (NfL), with evidence indicating that it may appear even earlier than the NfL marker (Irwin et al, 2024). A larger clinical enrollment to validate the reliability and sensitivity of its usage as a clinical biomarker is anticipated.

Besides mis-splicing caused by specific RBPs, defects of the general splicing machinery are also observed in neurodegenerative diseases (Fig. 2). Deficiency of the survival motor neuron (SMN) protein by genetic lesions in the *SMN1* gene is the cause of spinal muscular atrophy (SMA), a motor neuron degenerative disease (Lefebvre et al, 1995). SMN deficiency compromises the assembly of small nuclear ribonucleoproteins (snRNPs), the core components of the spliceosome (Battle et al, 2006), thereby resulting in widespread splicing defects in SMA (Lotti et al, 2012; Zhang et al, 2008). Additionally, FUS has been shown to interact with SMN, linking splicing defects in ALS and SMA (Mirra et al, 2017; Sun et al, 2015a; Yamazaki et al, 2012). In ALS caused by mutations in FUS/TLS (FUS-ALS), the cytosolic mislocalization of FUS disrupts the normal interaction between U1 snRNP and SMN complexes, which leads to perturbed snRNP assembly and RNA splicing in FUS-ALS (Jutzi et al, 2020; Sun et al, 2015a). Increased intron retention in many RBPs is observed in FUS mutant cells, which includes *FUS* mRNA itself (Humphrey et al, 2020; Luisier et al, 2018), forming a positive feedback loop in enhancing the splicing defects in FUS-ALS.

RNA splicing alterations also occur in AD. One identified reason is that U1 snRNP spliceosome components accumulate in the insoluble fraction, which perturbs the functionality of the spliceosome (Bai et al, 2013). In particular, the core subunit U1-70K was found to be cleaved to a N-terminal 40-KDa fragment (N40K), which exhibits a dominant-negative effect inhibiting the U1-70K function. This leads to mis-splicing and reduced expression of GABAergic synaptic genes, contributing to hyperexcitability in AD (Chen et al, 2022b). Furthermore, Tau aggregates were recently shown to induce cytoplasmic mislocalization and co-aggregation of nuclear speckle proteins, such as SRRM2 (Lester et al, 2021). Nuclear speckles are membraneless RNP granules enriched in the components of the RNA splicing machinery (Spector and Lamond, 2011). The cytosolic mislocalization leads to global splicing deficits, particularly increased intron retention. Alternative splicing can potentially impact various signaling pathways and thereby contribute to the molecular and cellular phenotypes of AD, including synaptic dysregulation, neuronal hyperexcitability, neuroinflammation, and chromosomal instability

(Chen et al, 2022b; Li et al, 2021). Overall, multiple molecular mechanisms underly the RNA splicing dysregulation in AD patients. Further elucidation of how each distinct pathway contributes to disease pathogenesis and how these different mechanisms influence each other requires additional studies.

Deficiency of spliceosomal function is also found in microsatellite repeat expansion diseases. The poly-dipeptide proteins derived from the *C9ORF72* microsatellite repeat expansion are able to block spliceosomal assembly by interacting with U2 snRNP (Yin et al, 2017). Furthermore, it was recently identified that the (GGGGCC)n repeat RNA co-localizes with nuclear speckles and affects its dynamic properties, and the poly-GR can induce SRRM2 cytoplasmic mislocalization and co-aggregation. The repeat RNA and dipeptide proteins synergistically lead to nuclear speckle dysfunction and global splicing deficits, most notably increased exon skipping and intron retention (Wu et al, 2024).

In summary, global RNA splicing dysregulation is broadly found in neurodegenerative diseases (Figs. 1 and 2), although the molecular mechanisms and the exact targets differ. In this review, we focused on examples highlighting that the perturbation of specific RBPs or spliceosomal machinery can trigger splicing defects, as this is more likely to be directly associated with pathogenic mechanisms for disease initiation and progression. The toxicity of RBPs (such as TDP-43 and FUS) usually includes both loss of function, due to nuclear clearance, and gain of toxicity arising from protein aggregation in the cytoplasm. Elucidating the distinct contributions to neurodegeneration is critical for designing effective therapeutic strategies. For TDP-43, increasing evidence suggests that nuclear clearance occurs before aggregation, indicating the significant contribution of splicing dysregulation in driving disease progression. The dysfunction of RBPs and the spliceosome can lead to changes in the expression patterns of hundreds or even thousands of genes. While it is likely that a few specific targets play pivotal roles in disease phenotypes, the possibility exists that the combined effects of multiple targets within the same pathway contribute synergistically to the overall outcome. Therefore, in addition to strategies aimed at correcting the mis-splicing of specific genes, it is equally important to explore approaches that restore the broader functional integrity of RBPs or spliceosomes, or that rescue cellular pathways enriched with defective genes.

## Alternative polyadenylation

Another mechanism that promotes transcript diversity is alternative polyadenylation (APA). Alternative isoforms with different lengths of 3'-UTRs can be produced via the use of APA sites (Tian and Manley, 2017). APA sites can also be found occasionally in intronic regions. Over 70% of mRNA-encoding genes exhibit APA isoforms (Tian and Manley, 2017). APA can regulate gene expression via different RNA processing pathways, including regulation of mRNA stability, nuclear export, translation, and subcellular localization. Aberrant use of APA sites can result in truncated mRNAs and abnormal protein expression (Passmore and Coller, 2022; Tian and Manley, 2017).

Widespread APA changes have been reported from large cohort studies of both C9ORF72-ALS and sporadic ALS brains (McKeever et al, 2023; Prudencio et al, 2015; Zeng et al, 2024), indicating a global trend toward distal 3'-UTR APA usage in ALS (McKeever et al, 2023) (Fig. 1). Though less abundant, intronic APAs also

show an increase in ALS (McKeever et al, 2023), which could potentially lead to truncated mRNAs, interfering with functional protein levels. Several ALS-related genes have isoforms with different lengths of 3'-UTRs due to APA, such as *TDP-43, MATR3, SETX, ANXA11*, and *TIA1* (McKeever et al, 2023). Through pathway enrichment, it is speculated that lengthened transcripts might influence organellar assembly and protein localization, whereas shortened transcripts might influence transcription and protein complex assembly (McKeever et al, 2023).

APA is known to be regulated by RBPs (Passmore and Coller, 2022; Tian and Manley, 2017). For example, knockdown of TDP-43 or disease-causative mutations of TDP-43 affect APA, which tends to favor the usage of distal sites (Arnold et al, 2024; Bryce-Smith et al, 2024; Polymenidou et al, 2011; Rot et al, 2017; Zeng et al, 2024). The extended 3'-UTR is considered to increase the stability of the transcripts, such as microtubule affinity-regulating kinase 3 (MARK3), the stabilization of which is suggested to increase the accumulation of phosphorylated tau (Arnold et al, 2024). APA and stabilization of multiple transcription factor-encoding transcripts could potentially have a broader influence on transcription (Bryce-Smith et al, 2024). Several disease-relevant genes, such as *ELP1, NEFL, TMEM106B*, were also reported to have longer 3'-UTRs upon TDP-43 reduction (Zeng et al, 2024). In addition, a recent study used single-nucleus RNA-sequencing in both familial and sporadic ALS, identified cell type-specific APA dysregulation in ALS, and applied a deep learning method to identify potential *cis/ trans* regulators of APA in disease (McKeever et al, 2023). The approach was validated by the identification of known regulators, such as TDP-43, and further revealed several splicing factors, including some with known APA functions in cancer, such as HNRNPC, SFPQ, and SRSF7. HNRNPC knockdown is known to promote distal polyadenylation site usage in cancer (Fischl et al, 2019), and its expression is downregulated in excitatory neurons of C9ORF72-ALS (McKeever et al, 2023). However, further investigation will be needed to confirm and further characterize the roles of those predicted RBPs on APAs in ALS.

Beyond ALS, APA is considered to potentially impact the function of risk genes in several neurodegenerative diseases. A recent study reported on a 3'-UTR APA transcriptome-wide genomic study in 11 brain disorders, and nominated a list of disease-associated APA-related genes, including *SNCA* in PD. It was found that 3'-extended usage of *SNCA* increases PD risk (Cui et al, 2023). RNA-seq analysis of data from AD, PD, and ALS suggests that genes with disease-specific dysregulation of APA are enriched in pathways related to protein turnover and mitochondrial function (Patel et al, 2019).

These studies affirmed an association of APA with pathological proteins and variants of neurodegenerative diseases. To enhance the understanding of APA, there is a need for improved bioinformatic tools, larger sample sizes, and functional studies investigating the relationships between APA, RBPs and disease variants. Moreover, the biological functions and pathological consequences of APA aberrations remain largely speculative, and thus need further experimental elucidation.

## RNA degradation

RNA degradation controls the steady-state pool of RNA levels at precise stage and location, and ensures the fidelity of RNA transcripts. When errors are introduced during transcription or RNA processing, different RNA-degradation pathways will be triggered to degrade the aberrant transcripts and avoid the production of defective proteins. For example, transcripts containing premature translation termination codons (PTCs) can be degraded by the nonsense-mediated decay (NMD) machinery (Daar and Maquat, 1988; Maquat, 2004). For transcripts lacking stop codons, which are usually caused by transcriptional errors, a Ski complex-mediated non-stop decay machinery will be involved (Frischmeyer et al, 2002; Garneau et al, 2007; van Hoof et al, 2002). When translation stalls on RNA transcripts, a no-go decay machinery will be engaged where binding of Pelota and HBS1L near the stalling sites will recruit the exosome and Xrn1 for endonucleolytic cleavage and decay (Doma and Parker, 2006; Garneau et al, 2007; Ikeuchi et al, 2016).

Abnormal global RNA stability has been reported in several neurodegenerative diseases (Figs. 1 and 3). A trend of mRNA destabilization is recognized in ALS fibroblasts and induced pluripotent stem cells (iPSCs) of C9ORF72-ALS (Tank et al, 2018). Accumulation of TDP-43 was proposed to account for the mRNA destabilization in the C9ORF72-ALS iPSC model, as a considerable proportion of destabilized transcripts are shared between iPSCs with overexpression of TDP-43 and iPSCs of C9ORF72-ALS (Tank et al, 2018). However, whether the iPSC model of C9ORF72-ALS exhibits TDP-43 overexpression, and whether the mechanism is conserved in neurons is unclear. A recent publication reported globally stabilized mRNAs in C9ORF72-ALS/FTD patient iPSC-derived neurons (iPSNs) and postmortem brain tissues (Li et al, 2023). It was found that the m⁶A RNA modification reduction in patient iPSNs mediates the decreased decay of m⁶A-marked transcripts, which are enriched in neuronal functions and synaptic activity (Li et al, 2023). Elevated accumulation of neuronal transcripts may potentially contribute to the hyperexcitability of ALS neurons.

Dysfunction of the NMD pathway has been implicated in several neurodegenerative diseases (Fig. 3). Overexpression of UPF1, an RNA helicase critical for NMD (Leeds et al, 1991), has been shown to have beneficial effects on reducing the toxicities induced by FUS, TDP-43, and *C9ORF72* repeat expansion, though the exact mechanisms vary (Barmada et al, 2015; Jackson et al, 2015; Ju et al, 2011; Ortega et al, 2020; Sun et al, 2020; Xu et al, 2019; Zaepfel et al, 2021). In cells with FUS and TDP-43 mutation and aggregation, UPF1 overexpression enhances the degradation of mis-spliced transcripts, including cryptic exons induced by TDP-43 loss of function, thereby reducing the potential neurotoxicity (Barmada et al, 2015; Kamelgarn et al, 2018). Overexpression studies suggest that UPF1 was insufficient in the diseased neurons; however, direct assessment of UPF1 expression has yielded controversial results. In C9ORF72-ALS/FTD, hyperactivation, inhibition or no changes in the NMD pathway have all been reported, and the protective role of UPF1 has been proposed to be exerted through NMD-mediated or NMD-independent mechanisms (Ortega et al, 2020; Sun et al, 2020; Zaepfel et al, 2021). Future studies comparing different models/cell types, and using an increased number of patient samples will help clarify these discrepancies. In addition, a recent study shows that UPF1 and UPF2 are both required for the degradation of some of the TDP-43-dependent PTC-containing mRNAs, suggesting a more complicated NMD mechanism with potentially compensatory components in the pathway (Alessandrini et al, 2024).

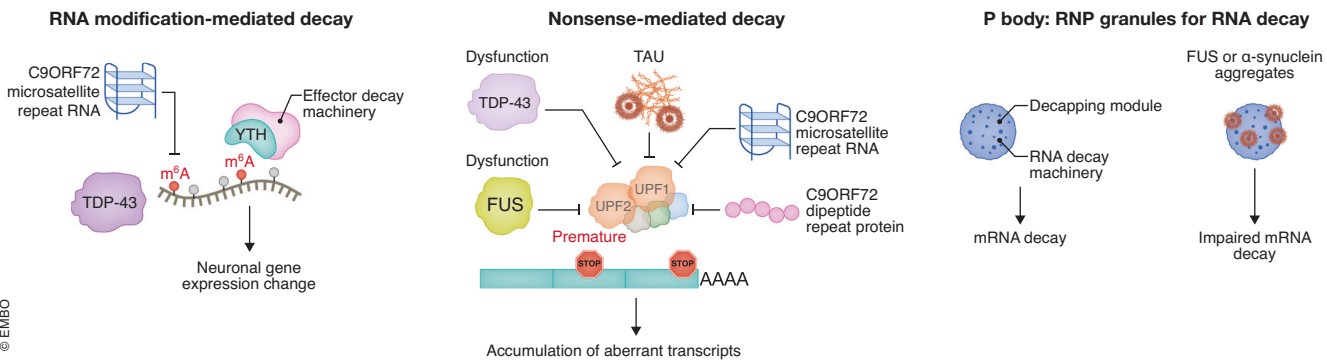

**Figure 3.   Impaired RNA decay mechanisms in different disease contexts.**

RNA decay can be disrupted via dysregulated RNA modifications, perturbed nonsense-mediated decay pathways, and dysfunction of decay-related RNP granules, such as P-bodies.

In addition, the membraneless RNP granules are also important for RNA degradation. Processing bodies (P-bodies) are cytoplasmic RNP granules primarily composed of components of the mRNA decay machinery and translationally repressed mRNAs (Fig. 3). It is generally believed that mRNAs recruited to P-bodies are destined for translational repression and degradation (Blake et al, 2024; Decker and Parker, 2012). The assembly and function of P-bodies have been shown to be perturbed in some neurodegenerative diseases. PD-causative mutation and cytoplasmic aggregation of alpha-synuclein directly modulate P-bodies by binding to multiple proteins in the P-body decapping module (e.g., DCP1/2, XRN1, and EDC3) on the EDC4 scaffold (Hallacli et al, 2022). Pathological alpha-synuclein in human neurons perturbs decapping module composition in P-bodies and disrupts mRNA decay kinetics, resulting in global stabilization of mRNAs (Hallacli et al, 2022). FUS does not localize to P-bodies, but it has been found that overexpression of ALS-causative mutant FUS can reduce the number of P-bodies (Takanashi and Yamaguchi, 2014). However, the RNA decay rate was not directly measured in that study (Takanashi and Yamaguchi, 2014); therefore, it remains unknown whether changes in P-body number due to FUS mutation functionally influence RNA decay.

In general, RNA degradation dysregulation in neurogenerative diseases is less extensively studied compared to other RNA processing pathways. The molecular mechanisms underlying these abnormalities often remain unclear, and many mRNA decay pathways have been rarely investigated beyond NMD. It is also noted that alterations in RNA stability under disease conditions may differ among cell types and/or vary across specific substrates. Future studies are needed to unravel the dysregulation mechanisms of different RNA decay pathways and further elucidate their roles in neuronal dysfunction and degeneration.

## RNA translation

RNA translation completes the life cycle of an mRNA, yielding the protein for the ultimate function. The efficiency of RNA translation can be influenced by intrinsic RNA structure, RBPs, translation factors, and cell signaling pathways. Translation of RNA in the context of neurodegenerative diseases with a focus on ALS has been recently reviewed elsewhere (Wang and Sun, 2023).

Global translation inhibition is generally found in AD, PD, ALS, FTD, and HD (Wang and Sun, 2023) (Fig. 1). Interaction of disease-linked proteins with core components of the mRNA translation machinery is a common mechanism. In C9ORF72-ALS/FTD, arginine-containing dipeptide repeat proteins (R-DPRs), poly-GR and poly-PR, bind to ribosomal subunits, inhibiting global protein synthesis (Hartmann et al, 2018; Kanekura et al, 2016; Loveland et al, 2022; Moens et al, 2019; Zhang et al, 2018b). It was shown by cryogenic electron microscopy (cryo-EM) that poly-GR and poly-PR block the polypeptide tunnel and the peptidyl transferase center, thereby inhibiting protein synthesis (Loveland et al, 2022). R-DPRs have also been shown to bind ribosomal RNA with a predicted affinity stronger than their binding to ribosomal proteins (Ortega et al, 2023). Furthermore, it was recently discovered that poly-GR perturbs translation elongation, increases ribosome collisions, and induces ribotoxic stress response, contributing to neurodegeneration of C9ORF72-ALS/FTD neurons (Dong et al, 2024). In HD, the aggregation-prone polyQ-expanded Htt protein shows stronger inter-action with ribosomal proteins than its soluble wild-type counterpart, inducing dose-dependent inhibition of global translation that is likely to result from a combination of defective ribosome biogenesis and ribosomal stalling/collision during elongation (Aviner et al, 2024; Culver et al, 2012; Eshraghi et al, 2021; Kim et al, 2016). Another neurodegeneration-associated protein, SMN, has been reported to prime ribosomes to a considerable subset of mRNAs, regulating their translation initiation. In SMA with loss of SMN function, ribosomes are depleted from the specific subset of mRNAs, resulting in inhibited translation (Bernabo et al, 2017; Lauria et al, 2020).

Global translation inhibition due to the activation of the integrated stress response (ISR) is widely found in neurodegenera-tive diseases (Storkebaum et al, 2023; Wang and Sun, 2023). The ISR signaling pathway allows cells to alter the protein expression to cope with cellular stress. It can be activated by several neurodegeneration-associated types of cellular stress such as oxidative stress, endoplasmic reticulum stress, proteotoxic stress, and neuroinflammation (Wang and Sun, 2023). Phosphorylation of eukaryotic translation initiation factor eIF2α is a signature of ISR

activation. This phosphorylation can be mediated by five eIF2α kinases: PERK, GCN2, PKR, HRI, and the recently identified MARK2 (Chen et al, 1991; Dever et al, 1992; Harding et al, 1999; Lu et al, 2021; Prostko et al, 1995; Storkebaum et al, 2023; Wang and Sun, 2023). Phosphorylation of eIF2α is elevated in AD patients, and suppression of the upstream kinases alleviates plasticity and cognition deficits in AD mouse models (Devi and Ohno, 2014; Lourenco et al, 2013; Ma et al, 2013; Segev et al, 2015; Tible et al, 2019). In C9ORF72-ALS/FTD, phosphorylation of eIF2α is also involved in the repeat-associated non-AUG translation of the microsatellite repeat expansion (Cheng et al, 2018; Green et al, 2017; Westergard et al, 2019). The kinase PKR and PERK have been suggested to be activated by the repeat RNA or poly-dipeptides (Zhang et al, 2014; Zu et al, 2020). In superoxide dismutase 1 (SOD1)-associated ALS, the mutant SOD1 protein is misfolded and forms aggregates, which trigger the unfolded protein response and endoplasmic reticulum stress, leading to PERK and PKR-mediated ISRs (Lindberg et al, 2005; Nishitoh et al, 2008; Saxena et al, 2009; Sun et al, 2015b). MARK2 is a newly identified kinase that mediates eIF2α phosphorylation independently of all the other kinases in response to proteotoxic stress, such as protein misfolding. It was found that G85R and A4V mutations of SOD1 can activate MARK2-mediated ISRs, in addition to the PERK-mediated ISRs (Lu et al, 2021). In FUS-ALS, phosphorylation of eIF2α is elevated in axons, suggesting local activation of ISR with early FUS pathologies (Lopez-Erauskin et al, 2018).

Global translation inhibition is a well-characterized feature of several neurodegenerative diseases. The fact that the same disease-causative RBPs and mutations lead to both global translation inhibition and RNA decay inefficiency causes a highly imbalanced stoichiometry between proteins and mRNAs. In addition, immediate ISR protects neuronal cells in response to stress, while persistent activation of ISR is harmful in neurodegenerative diseases. Future studies should investigate whether cell type specificity exists in the ISR persistence, and if inhibiting ISR could abrogate neurotoxicity in neurodegenerative diseases.

## Noncoding RNAs

In addition to protein-coding mRNAs, noncoding RNAs have also emerged as a relevant and evolving area of study in neurodegenerative diseases (Fig. 1). Noncoding RNA (ncRNA) constitutes a large and diverse domain, representing one of the major discoveries in the post-genomic era. The draft annotation of the human genome assembly T2T-CHM13 suggested the presence of over 43,000 genes of ncRNAs (>147,000 transcripts), which is about twice as many as the protein-coding genes/transcripts (Nurk et al, 2022). Loss-of-function studies have provided profound insights into the roles of ncRNAs in brain and neuronal function (Fatica and Bozzoni, 2014; Liu et al, 2017b; Sauvageau et al, 2013; Yang et al, 2021). Those ncRNAs include long noncoding RNAs (lncRNAs), microRNAs (miRNAs), circular RNAs (circRNAs), ribosomal RNAs, transfer RNAs, and many more. Here, we will focus on lncRNAs, miRNAs, and circRNAs because they have predominant regulatory functions and have been suggested to have potential functions in neurodegenerative diseases.

### Long noncoding RNAs (lncRNAs)

One feature of lncRNAs is that their expression, especially that of the long intergenic noncoding RNAs (lincRNAs), is strikingly tissue-specific, even more so than that of protein-coding mRNAs (Cabili et al, 2011). Currently, most of the lncRNA studies in the nervous system have focused on diseases related to neurodevelopment. Our understanding of the lncRNA mechanisms in neurodegenerative diseases is relatively more limited, and whether the function of lncRNAs in disease is deleterious or protective is controversial. Some well-documented examples of lncRNAs in the context of neurodegenerative diseases are nuclear paraspeckle assembly transcript 1 (NEAT1), Homeobox transcript antisense RNA (HOTAIR), and MALAT1.

NEAT1 is one of the most abundant lncRNAs in the nucleus, essential in paraspeckle formation (An et al, 2018). In PD, elevated NEAT1 levels have been reported in postmortem brains (Kraus et al, 2017; Simchovitz et al, 2019) and peripheral blood cells of PD patients (Boros et al, 2020). Multiple functions of NEAT1 elevation have been proposed. NEAT1 was found to stabilize the disease-causing PTEN-induced putative kinase 1 (PINK1), and to increase the accumulation in the mitochondrial membrane compartment of PINK1 that triggers autophagy (Pickrell and Youle, 2015). NEAT1 upregulation also promoted the expression of another PD-causing gene, SNCA, resulting in apoptosis (Liu and Lu, 2018). In addition, NEAT1 upregulation led to neuroinflammation through a number of pathways engaging the sponging of multiple microRNAs (Boros et al, 2021). In ALS and FTD, NEAT1 was found to associate with observed paraspeckle alterations in ALS/FTD patients with TDP-43 and FUS pathologies (Wang et al, 2020).

HOTAIR is a trans-acting lncRNA engaging in chromatin remodeling and epigenetic regulation (Raju et al, 2023). Increased levels of HOTAIR have been found in in vitro and in vivo models of PD. HOTAIR increase led to decreased expression of miR-221-3p and upregulation of the microRNA targets NPTX2 and α-synuclein, triggering the secretion of inflammatory cytokines and degeneration of dopaminergic neurons (Lang et al, 2020; Sun et al, 2022). A study also suggested a direct link between HOTAIR and the stabilization of the mRNA of a key pathological gene of PD, LRRK2 (Wang et al, 2017).

MALAT1 is a highly expressed lncRNA, localized at nuclear speckles (Arun et al, 2020). MALAT1 is increased in PD iPSNs (Abrishamdar et al, 2022). It was demonstrated that MALAT1 upregulation stabilizes α-synuclein and can increase SNCA level (Xia et al, 2019; Zhang et al, 2016a).

### microRNAs

miRNAs are evolutionarily conserved short RNAs with ~22 nucleotides in length, encoded by DNA stretches in either intergenic or intragenic regions (Gebert and MacRae, 2019). miRNAs can repress target gene expression by affecting either mRNA degradation or translational repression (Fabian et al, 2010; Gebert and MacRae, 2019). Even though their functions have been more extensively characterized in development, miRNAs are also important in regulating gene expression for neuron function in adulthood (McNeill and Van Vactor, 2012).

miR-7 is highly expressed in the brain (Farh et al, 2005). miR-7 binds to the 3'-UTR of α-synuclein mRNA. It is decreased in the substantia nigra of PD patients and of both in vivo and in vitro models of PD (Junn et al, 2009; McMillan et al, 2017). Loss of miR-7 leads to increased expression of α-synuclein, correlating with a loss of nigral dopaminergic neurons (McMillan et al, 2017). Introducing miR-7 to an MPTP-induced neurotoxin cell culture

model of PD, a commonly used PD model due to the selective effect of MPTP on dopaminergic neurons (Olanow and Tatton, 1999; Onofrj and Ghilardi, 1990), leads to downregulation of α-synuclein expression and shows a protective effect against oxidative stress (Junn et al, 2009; Li et al, 2016).

miR-133b was identified by a miRNA profiling analysis comparing the midbrains of PD patients and normal controls. This miRNA is specifically expressed in the dopaminergic neurons of midbrains and is drastically downregulated in PD patients. Overexpression of miR-133b in dopaminergic neurons suppresses maturation and dopamine release (Kim et al, 2007). Conversely, it was elevated in the plasma of a large cohort of PD patients and controls (Chen et al, 2021). The discrepancy between these studies could be attributable to the cell-type specificity of miRNA. In addition, the observed downregulation could result from the loss of dopaminergic neurons in the patients' brains. Therefore, further functional investigations will be needed to elucidate the role of miR-133b in PD.

miR-128 is expressed in adult neurons. Mice deficient in miR-128 develop neuroexcitability and fatal epilepsy due to miR-128 effects on the regulation of numerous genes involving ion channels, transporters, and neurotransmission (Tan et al, 2013). Over-expression of miR-128 leads to reduced motor activity and alleviation of motor abnormalities associated with PD-like disease and seizures in mice (Tan et al, 2013). The effects of miR-128 on neuroexcitability are likely through the regulation of various ion channels and the ERK2 signaling pathway (Tan et al, 2013). miR-128 downregulation has also been reported in HD patient brains and HD models (Kocerha et al, 2014; Lee et al, 2011; Marti et al, 2010). Targets of miR-128 include the key pathological HD gene *Htt*, and its regulators such as *HIP1* and SP1 (Kocerha et al, 2014).

### Circular RNAs (circRNAs)

Circular RNA (circRNA) has been extensively studied in the last few years and is increasingly linked to various neurodegenerative diseases. CircRNAs are products of back-splicing events, wherein the down-stream 5′ splice site of a precursor mRNA is ligated to the upstream 3′ splice site by a 3′-5′ phosphodiester bond, forming an RNA circle (Li et al, 2018). Alternatively, circular RNAs can also be generated from intron lariats that somehow escape debranching and degradation (Lasda and Parker, 2014). These circular intronic RNAs (ciRNAs) are generally less abundant than the exon-derived ones and relatively less well-studied. Over 11,000 circRNAs are expressed in the human brain (Dong et al, 2023). They are highly expressed in the nervous system and exhibit age-dependent accumulation, which could be partially due to their structural resistance to exonucleases (Gruner et al, 2016; Kim et al, 2021; Westholm et al, 2014). CircRNAs can cause changes in chromatin, transcription, splicing, and expression of their cognate linear mRNAs, as well as serve as sponges for microRNAs (Li et al, 2018; Meng et al, 2017).

Metadata generated from the Knight Alzheimer Disease Research Center (Knight ADRC) allowed the identification of 164 cortical circRNAs with significant association with AD traits, including AD diagnosis, pathologies, and co-expression with AD genes (Dube et al, 2019). The changes in circRNA levels differ from the expression changes of their cognate linear mRNAs. Among them, several circRNAs contain binding sites for microRNAs. For example, circHOMER1, the levels of which decrease with increasing dementia severity, possesses five predicted binding sites for miR-

651, whose downstream targets include the AD-related genes PSEN1 and PSEN2, suggesting a potential sponging mechanism of this circRNA (Dube et al, 2019). The BRAINcode project revealed that 29% of Parkinson's and 12% of Alzheimer's disease-associated genes produce circRNAs (Dong et al, 2023). Studies of specific circRNAs, such as circPSEN1, showed differential expression in autosomal-dominant individuals caused by pathogenic mutations in *APP, PSEN1*, and *PSEN2* (Chen et al, 2022a).

In ALS, FUS regulates the biogenesis of circRNAs by binding near the back-splicing junctions (Errichelli et al, 2017). In HD, the CAG microsatellite repeat expansion has been found to suppress circRNA biogenesis in mouse neural progenitor cells (Ayyildiz et al, 2023). Future studies should investigate whether circRNA levels are reduced in HD patients. Moreover, a study suggested the potential of using circRNA species as biomarkers for ALS in blood samples (Dolinar et al, 2019), which needs further validation.

The factors contributing to the changes in circRNAs in these neurodegenerative diseases have not been fully elucidated. One hypothesis is that the age-associated circRNA expression arises from a compromised splicing machinery, which suggests an interplay between different disease-related molecular pathways. This is supported by evidence indicating that depletion of spliceosomal components or treatment with splicing inhibitors can increase circRNA biogenesis from back-splicing events (Liang et al, 2017a). Alternatively, a recent study reported that increased transcriptional elongation speed (RNA polymerase II speed) is associated with elevated formation of circRNAs during aging (Debes et al, 2023), which is consistent with a previous study on the correlation between parental gene transcription elongation and circRNA back-splicing (Zhang et al, 2016b).

Despite substantial progress in characterizing the potential functions of ncRNAs in neurodegenerative diseases, our under-standing is still limited. Building on the observed correlations, it is important to understand how dysregulated ncRNAs contribute to the phenotypes of the diseases. Further functional studies will serve as crucial missing pieces of evidence. As noted in some examples above, ncRNAs have high cell specificity. Therefore, cell types should be taken into consideration when pursuing functional studies. Implementation of single-cell sequencing or spatial transcriptomic approaches could help understand the cell type distinction and spatial molecular interactions of ncRNAs in normal and disease contexts.

## RNA modifications

Over 160 chemical modifications have been identified in RNA molecules to date (Boccaletto et al, 2022). Any RNA molecule possesses at least one modification at some point during its life cycle. RNA modifications can impact almost all stages of the RNA life cycle. Chemical modifications on single nucleotides can alter the electrostatic charges of the RNA molecules, potentially influencing their phase separation properties. Ribosomal RNA (rRNA) and transfer RNA (tRNA) are thought to be the most heavily modified RNAs. Such modifications are usually not reversible, and they are critical for the RNA structures (Sloan et al, 2017). There has been an increasing number of different types of modification found in mRNAs as well as in regulatory noncoding RNAs. The most common ones on purine or pyrimidine bases are methylation, pseudouridylation, and adenosine-to-inosine (A-to-I)

editing (Delaunay et al, 2023). These modifications are reversible and play important roles in various steps of RNA processing regulation (Roundtree et al, 2017). Lately, pathological aberrant RNA modifications have emerged as critically involved in neurodegenerative diseases (Fig. 1).

$N^6$-methyladenosine (m⁶A) is the most prevalent internal mRNA modification in eukaryotic mRNAs (Dominissini et al, 2012). It is one of the few reversible RNA modifications that regulates RNA metabolism and abundance in the nervous system (Fan et al, 2023; Roundtree et al, 2017). m⁶A is installed by the "writer" methyltransferase complex composed of the core subunits METTL3 and METTL14, and it can be removed by the "eraser" protein FTO or ALKBH5 demethylase (Dominissini et al, 2012). "Reader" proteins selectively recognize m⁶A-marked RNA and determine the fate of these RNAs. A number of studies have demonstrated the important roles of m⁶A in regulating brain function, from development and synaptic plasticity to learning, memory, and neurodegeneration (Livneh et al, 2020; Wang et al, 2018). We recently found that m⁶A is globally reduced in poly-A RNAs in C9ORF72-ALS/FTD iPSC-neurons and postmortem brains, resulting in transcriptome-wide mRNA stabilization and gene expression elevation in the patient iPSC-derived neurons and postmortem motor cortex, particularly for genes involved in synaptic activity and neuronal function (Li et al, 2023). Moreover, m⁶A modification upstream of *C9ORF72* repeat expansion regulates the decay of both sense and antisense repeat RNA. Rescuing m⁶A modification in the patient neurons alleviated pathologies and improved neuron survival (Li et al, 2023). TDP-43 was reported to bind to m⁶A-marked RNAs (McMillan et al, 2023). Knockout of m⁶A reader YTHDF2 mitigates neurotoxicity in TDP-43-overexpressing primary neurons (McMillan et al, 2023). In sporadic ALS, both downregulation and upregulation of m⁶A modification levels have been reported in different reports. These studies used immunohistochemistry approaches to evaluate the global m⁶A signal, which cannot distinguish the contributions from structural RNAs (such as rRNAs) and mRNAs (Martin et al, 2022; McMillan et al, 2023). The discrepancy in the results suggests that either this staining approach is not quantitative, or that there is high heterogeneity among sporadic cases. It is possible that there are sub-groups of sporadic ALS cases that show different m⁶A changes induced by distinct molecular mechanisms. This requires further studies with larger numbers of controls and disease samples to be clarified.

The m⁶A methyltransferase METTL3 was reported to be reduced in AD brains (Castro-Hernandez et al, 2023; Huang et al, 2020; Zhao et al, 2021), which mirrored observations from primary cortical neurons treated with soluble Aβ oligomers (Zhao et al, 2021). Moreover, METTL3 knockdown in the mouse hippocampus induced oxidative stress, DNA damage, spine loss and neurodegeneration, and mouse cognitive behavior deficits (Zhao et al, 2021). Another study found that oligomeric tau induced the cytoplasmic translocation of the m⁶A reader protein hnRNPA2B1 and the cytoplasmic mislocalization of m⁶A (Jiang et al, 2021). Global downregulation of m⁶A was also reported in a PD-related cell model and rat brains. Reduction of m⁶A increased the expression of N-methyl-d-aspartate (NMDA) receptor 1, elevated calcium influx, and contributed to dopaminergic neurodegeneration (Chen et al, 2019). Moreover, a recent study found that abnormally expressed transcripts in the HD mouse brain had

reduced m⁶A levels, particularly adjacent to TDP-43-binding sites, which likely contributes to the alternative splicing changes in HD (Nguyen et al, 2023).

$N^1$-methyladenosine (m¹A) is highly abundant in structural RNAs, and present at low levels in mRNAs, where it is less abundant than m⁶A. m¹A is usually near the translation starting site (Dominissini et al, 2016). A recent study identified m¹A modification in the CAG repeat RNA, and its binding to TDP-43 contributes to the cytoplasmic mislocalization and aggregation of TDP-43 (Sun et al, 2023).

Pseudouridine (Ψ), an isomer of uridine, is present in noncoding RNAs (i.e., tRNA, rRNA, and snRNA) and in mRNAs (Schwartz et al, 2014). Unlike m⁶A, which is enriched at the stop codon of mRNAs, pseudouridines are distributed along the whole length of the mRNAs from the 5′- to the 3′-UTR without any region-specific enrichment (Khoddami et al, 2019; Schwartz et al, 2014). The functions of pseudouridine include stabilizing RNA structures and destabilizing interactions with RBPs (reviewed in (Borchardt et al, 2020)). In myotonic dystrophy type 2 (DM2), a neuromuscular disease involving neuronal loss and global neuronal impairment, pseudouridines within the CCUG repeats in the intron of *CNBP* decrease repeat RNA dynamics and thus reduce sequestration of MBNL1 to the repeat RNA (deLorimier et al, 2017). The link between pseudouridines and neurodegenerative diseases has not been extensively studied. It was demonstrated that acute oxidative stress in cells could trigger a significant elevation of pseudouridine levels in mRNAs (Feiler et al, 2015). It would be interesting to examine pseudouridine changes during aging or under specific pathological conditions.

A-to-I editing converts adenosines to inosines in both coding and noncoding RNAs. It is catalyzed by adenosine deaminases acting on RNA (ADARs), which are found to be enriched in the nervous system. A-to-I editing can affect base pairing, alter codons, and change splice sites. A recent computational trait-association study suggested that ADAR-mediated A-to-I editing in double-stranded regulatory RNA may underlie multiple neurodegenerative diseases, including AD, PD, and ALS (Li et al, 2022). An editing defect of the *GluR2* mRNA has been found in ALS spinal motor neurons due to the downregulation of ADAR2 (Hideyama et al, 2012; Kawahara et al, 2004; Kawahara et al, 2003; Takuma et al, 1999). The reduced editing at the Q/R site of GluR2 leads to increased calcium influx and excitotoxicity to motor neurons, eventually triggering neuronal death (Kawahara et al, 2004; Kawahara et al, 2003; Takuma et al, 1999). Moreover, the increased calcium influx activates the calcium-dependent cysteine protease calpain, which cleaves TDP-43 and produces C-terminal fragments that mis-localize and aggregate in the cytoplasm (Yamashita et al, 2012). In addition, ADAR2 is reported to be mislocalized in C9ORF72-ALS/FTD, leading to widespread RNA editing aberrations, especially in the pathways of integrated stress response and EIF2 signaling (Moore et al, 2019). The Q/R editing of *GluR2* mRNA was also found to be reduced in the hippocampus of AD patients (Akbarian et al, 1995; Gaisler-Salomon et al, 2014; Khermesh et al, 2016), which was recently suggested to potentially function as an epigenetic switch that regulates dendritic spines and links to neurodegeneration and memory deficits in AD (Wright et al, 2023). In synucleinopathy-associated neurodegenerative diseases, such as PD and dementia with Lewy bodies (DLB), the expression of the editing inhibitor ADAR3 is reduced by

transcription inhibition due to the insolubility of NONO/SFPQ, thereby increasing the editing on many transcripts encoding axonal, synaptic and mitochondrial proteins. These aberrantly edited RNAs are retained in the nucleus, leading to reduced protein levels (Belur et al, 2024).

The understanding of RNA modifications in neurodegenerative diseases has been expanding rapidly in the past few years. Studies of RNA modifications, also referred to as epitranscriptomics, provide a valuable angle to study neurodegenerative diseases beyond the primary genetic code. As in the above examples, RNA modification changes can influence multiple steps of RNA metabolism, including RNA decay and RBP localization. In addition, due to the prevalence of some RNA modifications, the impact of such modifications is likely to be broad on the transcriptome. It coincides with the technological developments that enable the measurement of RNA modifications with increased resolution and accuracy. The diversity of different types of RNA modifications is increasingly recognized in neurodegenerative diseases. In addition to deciphering the relationship between individual RNA modifications and disease physiology and pathology, it will be equally important to understand the interactions among different RNA modifications in the future. Moreover, the key mechanisms driving age-related or disease-specific epitranscriptomic changes remain largely unexplored. Performing more basic research to address the role of RNA modifications during aging, in different brain regions and cell types, will be helpful to enhance our understanding of neurodegenerative diseases.

## Nucleocytoplasmic transport of RBPs

Many RBPs are predominantly localized in the nucleus, but also constantly shuttle between the nucleus and the cytoplasm, participating in multiple steps of RNA processing. However, it has been found in many neurodegenerative diseases that RBP proteins are mislocalized to the cytoplasm, which results in the loss of their nuclear functions (Fig. 4). As many of those RBPs are aggregation-prone, they can form cytosolic inclusions and exhibit gain of toxicity. Exploring the factors contributing to the mislocalization of RBPs is crucial for understanding disease mechanisms.

The nucleocytoplasmic transport of RBPs is bidirectional. Efficient nucleocytoplasmic transport requires functional nuclear pore complexes (NPC), a Ran gradient, and nuclear transport receptors. NPCs are located on the nuclear envelope. They are composed of over 30 nucleoporins (Nups), which are arranged in an eightfold rotational symmetry (reviewed in (Khan et al, 2020)). A nucleoplasmic-cytoplasmic gradient of RanGTP–RanGDP across the nuclear envelope, generating asymmetry between the nucleoplasm and the cytoplasm, provides essential directionality to nucleocytoplasmic transport (Nachury and Weis, 1999). Nuclear transport receptors include importins and exportins (Ding and Sepehrimanesh, 2021). Importins recognize nuclear localization signals (NLS) in RBPs to transport the RBPs from the cytoplasm to the nucleus. Some well-characterized importins include importin β1, transportin 1 and 3 (TNPO1/3). Exportins (XPOs) recognize nuclear export signals (NES) in RBPs to transport the RBPs from the nucleus to the cytoplasm (Ding and Sepehrimanesh, 2021). There is also a third class of transport receptors, named biportins, which are relatively ambiguously defined. Biportins can function bidirectionally as importins or as exportins (Yang et al, 2023). Some mutations in RBPs are located in their NLS, therefore affecting the nuclear import of the protein, such as in FUS NLS mutations (Kwiatkowski et al, 2009; Mackenzie et al, 2010). Wild-type RBPs can also be mislocalized in sporadic diseases, due to NPC/transport machinery defects.

Nuclear pore complex (NPC) components have been shown to be perturbed in ALS and FTD, as evidenced by morphological irregularities and altered expression of nucleoporins. This perturbation could contribute to nucleocytoplasmic mislocalization of RBPs, including TDP-43. Different mechanisms could contribute to such nuclear pore defects. First, NPCs are susceptible to aging (Sakuma and D'Angelo, 2017). In post-mitotic cells, NPCs—especially the NPC scaffold components—are extremely long-lived with a very slow turnover rate (D'Angelo et al, 2009; Savas et al, 2012). Given the long life of NPCs, age-related damage accumulation of NPCs eventually results in increases in nuclear permeability and abnormal distribution of nuclear and cytoplasmic proteins in aged neurons (D'Angelo et al, 2009). Second, specific mutations or pathologies can cause impairment of the nuclear pore complex. For example, the expression of C9ORF72 microsatellite repeat expansion RNA and proteins is associated with NPC defects (Coyne et al, 2021; Freibaum et al, 2015), partially due to increased nuclear expression and localization of CHMP7, an NPC quality control protein that impacts NPC homeostasis on the nuclear envelope and leads to TDP-43 leakage into the cytoplasm (Coyne et al, 2020).

In AD, pathological phospho-tau can directly interact with NPC components and trigger mislocalization of some Nups, thus disrupting the NPC function and compromising the NPC diffusion barrier (Eftekharzadeh et al, 2018; Lester and Parker, 2018). In HD, the expression of microsatellite CAG repeat expansions in the huntingtin (htt) gene results in morphological changes of NPCs and consequently mRNA nuclear retention (Gasset-Rosa et al, 2017). In addition, the aggregates formed by polyglutamine-rich proteins are found to co-aggregate with NPC components or trigger the mislocalization of NPC components to stress granules, which influences the solubility and functionality of NPCs (Gasset-Rosa et al, 2017; Grima et al, 2017; Shi et al, 2017; Zhang et al, 2018a). Moreover, the aberrant phase transition of RBPs, which is discussed in more detail in the next section, can also influence the NPC function. For example, it was found that the insoluble TDP-43 cytoplasmic inclusions could potentially sequester nucleoporins and transport factors, and even trigger the mislocalization of some of the nucleoporin proteins (Chou et al, 2018; Khalil et al, 2022).

Disrupted Ran gradient and nuclear transport receptors have also been reported in some neurodegenerative diseases. Huntingtin-linked polyglutamine induces cytoplasmic mislocalization of RanGAP1, a regulatory protein that hydrolyses Ran-GTP to Ran-GDP (Gasset-Rosa et al, 2017). Poly-GA is reported to impair the importin-α/β-dependent pathway and lead to compromised import of TDP-43 (Khosravi et al, 2017). In AD hippocampal neurons, both RanGDP transporter and importins have been reported to be mislocalized to the cytoplasm (Lee et al, 2006; Sheffield et al, 2006). The link between Ran gradient dysregulation and neurodegeneration is best-characterized in C9ORF72-ALS/FTD. The Ran gradient and its associated regulatory proteins such as RCC1 are disrupted in multiple C9ORF72-associated ALS/FTD models, including long differentiated iPSC-neurons, induced neurons of carriers, and Drosophila models expressing $G_4C_2$ repeats (Freibaum et al, 2015;

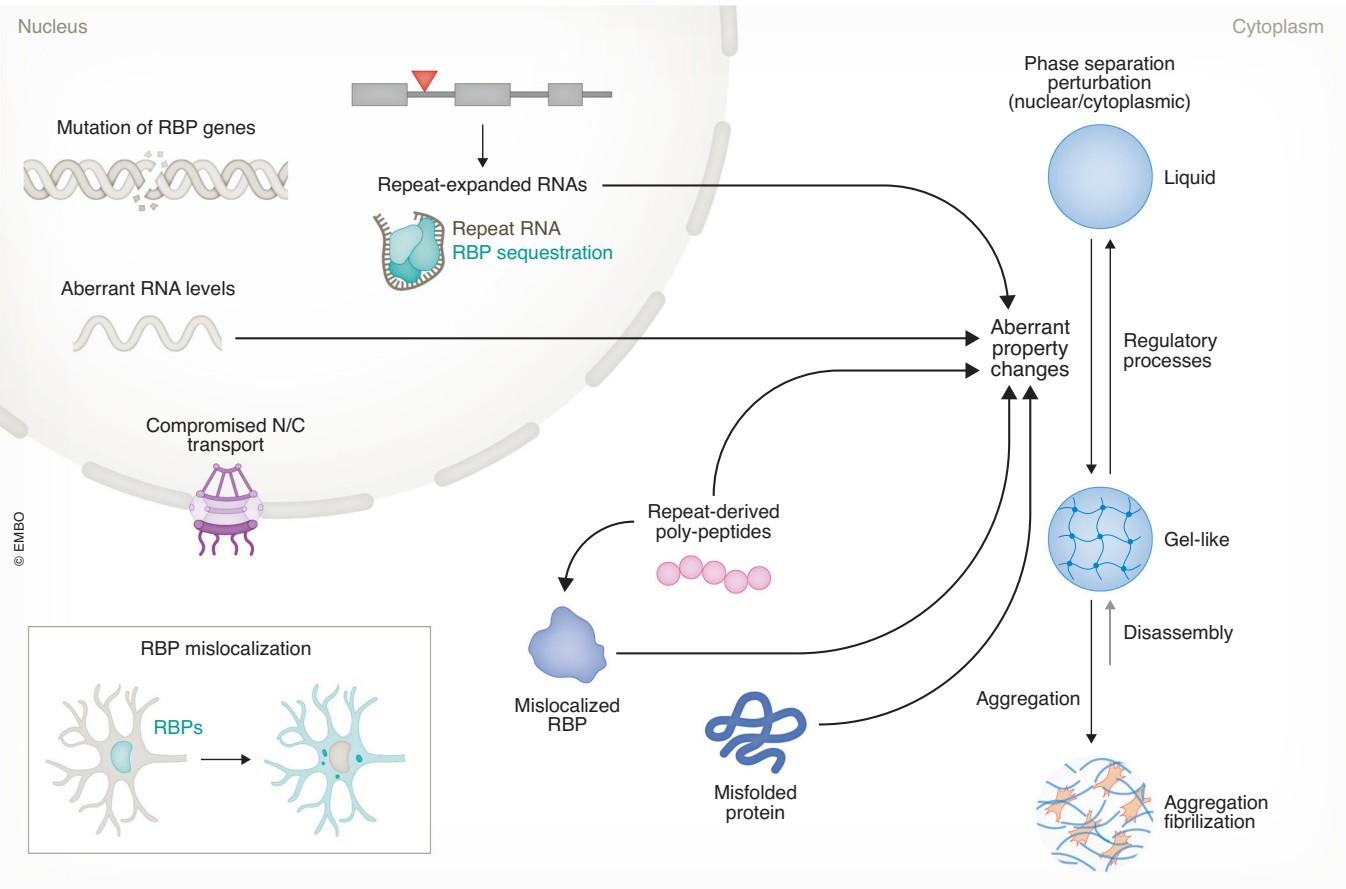

**Figure 4. Potential causes of RNA dysregulation in neurodegenerative diseases.**

RNA dysregulation is primarily driven by RBP dysfunction, which can be attributed to genetic mutations, abnormal expression levels, mislocalization, aberrant phase separation or aggregation, or perturbation by repeat expansions. Many RBPs are multifunctional and can impact a wide range of RNA processing pathways.

Jovicic et al, 2015; Klebe et al, 1995; Zetsche et al, 2015). Nuclear mislocalization of RanGAP1 has been observed in C9ORF72-ALS/FTD postmortem motor cortex, and overexpression of RanGAP1 rescued the impaired Ran gradient in patient-derived neurons (Zetsche et al, 2015). Antisense oligonucleotides (ASOs) targeting *C9ORF72* sense or antisense repeat RNAs have been shown to attenuate nucleocytoplasmic transport abnormalities and NPC defects linked to some of the Nups (Coyne et al, 2020; Rothstein et al, 2023; Zetsche et al, 2015).

In addition, other mechanisms may also contribute to RBP nucleocytoplasmic mislocalization. Given the RNA-binding nature of RBPs, the localization of RBPs can be influenced by the distribution of target RNAs. Acute depletion of RNA abundance in the nucleus by transcriptional inhibition of RNA polymerase II can induce FUS cytoplasmic mislocalization (Tsai et al, 2022), as well as TDP-43 mislocalization (Duan et al, 2022). The nuclear RNA binding of TDP-43 has been shown to be important for its nuclear retention (Duan et al, 2022). As mRNAs exhibit extensive nucleocytoplasmic distribution alterations in ALS and FTD (Fernandopulle et al, 2021; Kim et al, 2017; Markmiller et al, 2021; Tsai et al, 2022), this could potentially promote RBP mislocalization.

Deficiency of nucleocytoplasmic transport provides a plausible mechanism for RBP mislocalization and RNA metabolism dysregulation. However, many important questions remain unanswered. For example, current evidence of NPC defects appears to be circumstantial. Therefore, it could be of great importance to identify the exact causes of NPC defects in neurodegenerative diseases. Beyond current evidence of expression changes in NPC components, the topology and structure of NPCs should also be investigated to better understand their dysregulated functions in disease. Furthermore, comparing NPC structure and function across different cell types will help delineate the potential mechanisms underlying neuron-specific deficits. Nucleocytoplasmic transport is a coordinated process involving events between the nucleus and the cytoplasm, and identifying which interactions are essential or which events occur early in disease progression could provide useful insights for the development of potential therapeutic strategies.

## Phase separation of RBPs

Phase separation is a local concentration change of molecules that allows the formation of membraneless condensates (Lyon et al,

2021; McDonald et al, 2020; Molliex et al, 2015). Biomolecular condensate formation is a universal phenomenon in cells, playing essential roles in various cellular processes, such as transcription, translation, and signaling (Lyon et al, 2021; McDonald et al, 2020; Molliex et al, 2015). Various membraneless RBP granules are condensates formed with both proteins and RNAs. Many RBPs contain intrinsically disordered regions (IDRs) that facilitate phase separation.

Stress granules are an extensively studied type of phase-separated granules that show links to neurodegeneration. They are formed in response to cellular stress and are composed of non-translating mRNPs (Marcelo et al, 2021; Weskamp and Barmada, 2018). They are enriched with mRNAs, RBPs, and translation initiation factors, and depleted of deadenylation proteins. There-fore, they are largely considered to maintain mRNA stability (Marcelo et al, 2021; Weskamp and Barmada, 2018). However, the exact function of stress granules has not been completely elucidated. Stress granules are transient and dynamic structures. Persistent stress granule formation or delayed stress granule disassembly are considered to contribute to protein aggregation in neurodegenerative diseases (Marcelo et al, 2021; Wolozin, 2012). This hypothesis is supported by evidence indicating that many neurodegeneration-associated proteins are found in stress granules, including TDP-43, FUS, hnRNPA1, and Tau (Baron et al, 2013; Cruz et al, 2019; Gui et al, 2019; Li et al, 2013; Liu-Yesucevitz et al, 2010). However, stress granule marker proteins, such as G3BP1 and TIA1, are rarely found to co-aggregate with these pathological protein inclusions in patient postmortem tissues. One possibility is that stress granules can increase the propensity of specific RBPs, not the whole stress granules, to form aggregates. However, the mechanism and the connection of stress granules to RBP aggregation need further elucidation.

Alternatively, RBPs can form independent condensates by themselves. Posttranslational acetylation of TDP-43 has been found to drive the demixing of TDP-43 into intranuclear liquid crystal spherical shells with liquid cores, named "anisosomes" (Yu et al, 2021). Optogenetic TDP-43 nucleation induces insoluble aggre-gates, which do not co-localize with stress granules (Mann et al, 2019; Otte et al, 2020). FUS phase separation in cells has been found to be modulated by posttranslational methylation (Qamar et al, 2018). In vitro experiments have shown that droplets of FUS itself can convert into fibrous structures over time, and this process can be accelerated with patient-derived FUS mutations (Murthy et al, 2019; Patel et al, 2015; Tetter et al, 2024). Time-lapse imaging revealed the nucleation process of FUS forming short fibers to long fibers (Patel et al, 2015).

The aberrant phase separation of RBPs is considered to be an important mechanism of protein aggregation in neurodegenerative diseases (Nussbacher et al, 2019) (Fig. 4). Many disease-causing mutations tend to cluster in the low complexity domains (LCDs) with intrinsically disordered regions (IDRs) (Lagier-Tourenne et al, 2010), which are tightly associated with phase separation (discussed in more detail later). These mutation clusters are found in the C-terminal domains of TDP-43 (Lagier-Tourenne et al, 2010; Nussbacher et al, 2019), hnRNPs (Kim et al, 2013), and TIA1 (Mackenzie et al, 2017), as well as in the N-terminal domains of FET family proteins FUS (Kwiatkowski et al, 2009; Vance et al, 2009), EWS (Couthouis et al, 2012) and TAF15 (Neumann et al, 2011). Failure to maintain the liquid phase homeostasis can result

in RBP loss of function and influence a spectrum of protein homeostasis and cellular functions. Emerging evidence confirms that the phase separation process is a complex multifactorial transformation (Carey and Guo, 2022), with multivalence serving as a consensus mediator. The multivalence can arise from the IDRs of RBP proteins, the multivalent nature of RNAs, and versatile electrostatic interactions. In neurodegenerative diseases, multiple factors may contribute to aberrant phase separation and eventually lead to protein aggregation, such as changes in posttranslational modifications and protein homeostasis pathways during aging, mutations in IDRs that influence electrostatic interactions, and dysregulated RNAs resulting from impairment of RNA metabolism (Liu et al, 2017a; Nussbacher et al, 2019). Therefore, aberrant phase separation is likely a collective consequence of defects or changes in RBPs, RNAs, and the surrounding microenvironment. The changes in RNA metabolism and the mislocalization of RBPs discussed above may ultimately contribute to the aberrant phase separation that is widely observed in many neurodegenerative diseases, and such aberrant phase separation of RBPs could be further amplified and propagate under disease conditions.

It is now understood that proteins prone to undergo phase separation are characterized by several features, such as IDR domain, multivalency, and RNA engagement. However, there is still a lack of evidence linking phase separation to aggregation formation observed in patients. One current bottleneck in advancing this field further lies in the limited tools for monitoring and capturing in vivo phase separation and aberrant transition in neurons and neurodegenerative disease models. Recent advances in the available techniques for studying phase separation in vivo deepen our understanding of neurodegeneration-associated phase separation. Recently, several studies used optogenetic tools to visualize the nucleation process of FUS and TDP-43 in cells (Otte et al, 2020; Shimobayashi et al, 2021). Recent work using cryo-EM has revealed more precise structures and protein components of the aggregates in patient postmortem tissues, such as TAF15 (Tetter et al, 2024) and TDP-43 (Arseni et al, 2023; Arseni et al, 2024) that formed amyloid filaments in different types of FTD. Such technological advances can help elucidate the protein-protein and protein-RNA interactions, providing insights into the mechanisms of phase separation transition and RBP protein aggregation in neurodegenerative diseases.

## RNA repeat expansion and neurodegeneration

Microsatellite repeat expansion causes over 50 neurological and neuromuscular diseases, including spinocerebellar ataxia, myotonic dystrophy (DM1 and DM2), HD, and ALS/FTD (Fig. 4). RNAs with repeat expansion can induce various toxicities and contribute to disease pathogenesis, especially when the expansion is located in the noncoding regions of the gene, including the 5'- and the 3'-UTRs, and introns (Paulson, 2018). In many cases, the micro-satellite repeat-expanded sequence can be bidirectionally tran-scribed (Gudde et al, 2017; Haeusler et al, 2016; Paulson, 2018), although the mechanism remains unclear. Transcripts from both sense and antisense strands have been detected in C9ORF72-ALS/FTD, HD, SCA7, and DM1 (Castro et al, 2020). It is usually thought that the antisense transcripts may be produced at lower levels than the sense transcripts (Paulson, 2018), but based on evidence from C9ORF72-ALS/FTD, this may not always be true. However, there is

limited information on the stoichiometry of sense and antisense transcripts at the single-cell level.

The RNAs with repeat expansions can form RNA foci in the nucleus, regardless of sense or antisense origin (Paulson, 2018). The repeat RNA foci can sequester RBPs and cause their loss of function. A clear line of evidence on repeat RNAs sequestering RBPs and influencing RNA splicing refers to myotonic dystrophy type 1 and 2 (DM1/DM2) where CUG- or CCUG-repeat RNA binds MBNL1–3, muscle or neuron-enriched RBPs. Loss of MBNL function leads to global splicing and APA dysregulation of genes important for muscle or neuron development and function, contributing to the muscle and eye symptoms of patients (Fardaei et al, 2002; Kanadia et al, 2003; Lin et al, 2006; Mankodi et al, 2001; Miller et al, 2000). This mechanism has been confirmed in multiple disease models. MBNL depletion mirrors the majority of splicing defects in DM1 patients, and overexpression of MBNL significantly rescues the splicing deficits in the disease (Kanadia et al, 2003; Kanadia et al, 2006). In *C9ORF72* repeat expansion-linked ALS and FTD, the sense (GGGGCC)n and antisense (CCCCGG)n repeat expansion have also been shown to interact with RBPs. However, instead of one dominant RBP being sequestered by the repeat RNA, as seen in the DM1 case, multiple RBPs have been reported to bind to the C9ORF72 expanded repeats. These include Pur-α (Xu et al, 2013), RBPs involved in RNA transcription, editing, splicing and transport (Celona et al, 2017; Cooper-Knock et al, 2014; Donnelly et al, 2013; Haeusler et al, 2014), master regulator hnRNP family protein (Conlon et al, 2016; Cooper-Knock et al, 2014; Mori et al, 2013), and increasing paraspeckle components (Bajc Cesnik et al, 2019). More examples can be found in other reviews (Conlon and Manley, 2017; Nussbacher et al, 2019; Schwartz et al, 2021). A recent study revealed an alternative mechanism suggesting that, instead of sequestering one specific protein, the (GGGGCC)n repeat RNA affects the biophysical properties and functions of the endogenous nuclear speckles, a type of nuclear RNP granules, leading to global splicing repression (Wu et al, 2024).

Apart from the repeat RNA-mediated toxicity, the repeat peptides encoded by the RNA repeats can also show toxicity in neurons. The repeat RNAs, especially those with secondary structures, undergo non-canonical AUG-independent repeat-associated (RAN) translation in all possible reading frames. Multiple poly-peptides or poly-dipeptides can be generated from each repeat sequence, and RAN translation has been found for many repeat expansion sequences, including those in the coding region (Banez-Coronel and Ranum, 2019). The cellular toxicity of different peptide repeats varies, influenced by the amino acid composition (Freibaum and Taylor, 2017). The RAN translation can begin using the 5' cap scanning mechanism, and can also occur independently of the 5' cap, such as in intron-localized repeats (Wang and Sun, 2023). Many translation initiation factors and RBPs have been reported to regulate the translation efficiency of different repeats, which can influence neurotoxicity via modulating the toxic protein accumulation (Wang and Sun, 2023).

Translation elongation has also been found to be affected by some repeat sequences, especially arginine-containing peptide repeats that can be produced from GGGGCC repeats and CGG repeats (Park et al, 2021; Radwan et al, 2020; Wang and Sun, 2023; Wright et al, 2022). In C9ORF72-ALS/FTD, two-color single molecule imaging techniques were used to reveal that translation elongation of GGGGCC repeats is faster in the GA frame than in

GP, and the GR frame was characterized by the lowest elongation rate (Latallo et al, 2023). It has also been reported that GGGGCC repeats produce aggregation-prone chimeric DPR species containing GA and GP (McEachin et al, 2020). In fragile X-associated tremor/ataxia syndrome (CGG repeats) and SCA6 (TGGGCC repeats), chimeric DPR species due to translational frameshifts were also detected (McEachin et al, 2020; Wright et al, 2022). Several repeat proteins have been shown to affect various RNA processing pathways, such as translation and splicing, discussed in earlier sections.

Overall, the expression of genetic repeat expansions is a prominent cause of neurodegeneration, triggering cascades of multifaceted dysregulation. Both the repeat RNA and the generated peptide repeats can perturb multiple RNA processing pathways via different mechanisms (Fig. 4). It has been challenging to determine which repeat-derived product is the critical factor in disease pathogenesis, repeat RNA or peptide, sense or antisense, and which repeat peptide. It is possible that all of these factors contribute to disease, and there might be synergistic neurotoxic effects. In addition, different species may exhibit predominant toxicity in different cell types and at various disease stages. More studies are needed to fully understand these interactions.

## Biomarkers and therapeutics

Mechanistic studies have enabled the discovery of a range of promising potential biomarkers for early detection and stratification of patients. Currently established general molecular biomarkers for neurodegenerative diseases, mainly for AD and ALS, are neurofilament light chain and its phosphorylated heavy chain in serum and cerebrospinal fluid (CSF) (Loeffler et al, 2020; Rossi et al, 2018; Staffaroni et al, 2022; Verde et al, 2019). Aβ peptides, total tau level, phosphorylated tau level, and β-synuclein are biomarkers for AD (Alcolea et al, 2023).

Recently, research on RNA metabolism has been making significant contributions to biomarker development, particularly cryptic splicing targets of TDP-43. Two recent publications are promising regarding the detection of cryptic peptides in the CSF or iPSC-neurons of ALS/FTD patients (Irwin et al, 2024; Seddighi et al, 2024). Remarkably, the TDP-43-dependent cryptic epitope of *HDGFL2* was detectable in the CSF of pre-symptomatic C9ORF72-ALS/FTD carriers (Irwin et al, 2024), suggesting that CE and its cryptic peptides may facilitate earlier diagnosis of ALS. Further studies will be necessary to explore more CE events and cryptic peptides, develop sensitive and reproducible assays to detect CE RNA and/or peptides, compare or combine CE features with established neurofilament biomarkers, and investigate their longitudinal changes during disease progression. It is expected that the development of a panel of molecular biomarkers, monitoring different molecular pathways, could provide better guidance for clinical trial design and patient stratification.

The increasing understanding of RNA functions enhances the development of RNA-targeting therapeutic approaches in neurodegenerative diseases (Fig. 5). For diseases associated with dominant gain-of-function mutations, targeted RNA-silencing approaches, such as RNA interference (RNAi) and antisense oligonucleotides (ASO), are broadly used and some have been clinically validated and approved. Here, we will focus on ASO implication in neurodegenerative diseases. Thorough discussions

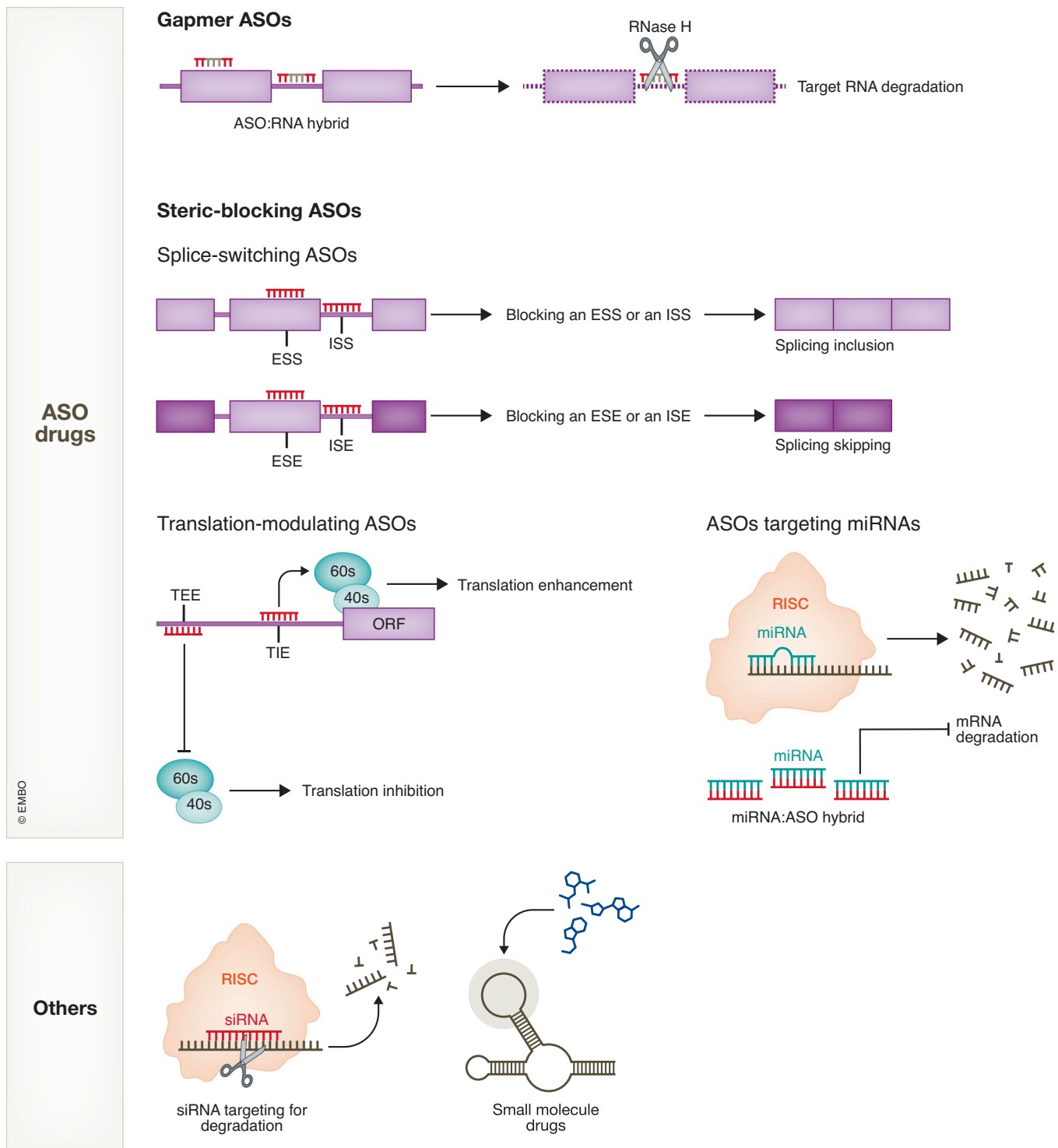

**Figure 5. Overview of RNA-targeting therapeutic approaches.**

Antisense oligonucleotide (ASO) drugs can be classified according to their modes of action. Gapmer ASOs bind to target mRNAs, recruiting endogenous RNase H for target mRNA degradation. Steric-blocking ASOs bind to target mRNAs without inducing degradation; instead, they modulate RNA processing, such as splicing and translation, by preventing RBP binding or altering the RNA structure. They can also base-pair with miRNAs to prevent them from targeting mRNAs for decay. ESS/ISS exonic/intronic splicing-suppressor element, ESE/ISE exonic/intronic splicing-enhancer element, TEE translation-enhancing element, TIE translation-inhibitory element. In addition to ASO drugs, other options for RNA-targeting therapies include siRNAs, which promote mRNA decay, and small-molecule drugs that target specific RNA structures to modulate their processing.

**Box 1    Questions and future directions**

Here are some key questions that are important to address and prioritize in order to better understand the role of RNA metabolism dysregulation in neurodegenerative diseases.

**Mechanisms linking RNA dysregulation to pathogenesis:**

How do specific RBPs and their loss of function or aggregation lead to neurotoxicity?
What are the critical RNA targets that mediate different aspects of neuronal dysfunction and degeneration?

**Cross-talk between RNA metabolism and other pathways:**

How does RNA dysregulation interact with other cellular dysfunctions, such as protein aggregation, mitochondrial deficits, stress signaling pathways, and neuroinflammation?
Does RNA dysregulation serve as a primary trigger or amplify downstream pathological cascades?

**Role of aberrant ncRNAs:**

Is dysregulation of ncRNAs linked to genetic variants in noncoding regions of the genome associated with neurodegenerative diseases?
How does dysregulation of ncRNAs exacerbate disease phenotypes?

**Selective vulnerability:**

What mechanisms underlie the aggregation, mislocalization, or dysregulation of RBPs in various neurodegenerative diseases?
Are there distinct proteinopathy patterns of RBPs across different neuronal subtypes and brain regions, or can the same RBP pathology predominantly lead to varied dysregulation of RNA processing pathways or targets, thereby contributing to cell type-specific vulnerability?
Are there protective mechanisms in resistant cell populations?

**Non-cell autonomous toxicity:**

Is there RBP dysfunction in glial cells, including astrocytes, oligodendrocytes and microglia, that contributes to disease progression in neurodegeneration?
How is RNA metabolism dysregulated in different glial cell types, and how does this contribute to their toxicity to neurons?

**Disease modeling:**

How can RBP proteinopathy, including aberrant aggregation, cleavage, modifications and mislocalization, be effectively modeled to mimic disease pathology in both in vitro and in vivo systems?
How can models be developed to study RNA dysfunctions and pathogenic mechanisms for sporadic diseases?

**Biomarkers and therapeutics:**

Can diagnostic or prognostic biomarkers be developed based on dysregulated RNA transcripts, to guide therapeutic design and precision medicine in neurodegenerative diseases?
Is targeting a specific RNA substrate sufficient to rescue disease phenotypes, or is restoring the overall function of the RBP or RNA processing pathway required?

---

about siRNA drugs and small-molecule compounds can be found in other reviews (Ahn et al, 2023; Angelbello et al, 2020; Perez-Arancibia et al, 2022; Setten et al, 2019).

ASOs have emerged as a promising RNA-targeting therapeutic approach. They are DNA oligos designed to target complementary RNA sequences via Watson–Crick base pairing (Levin, 2019). The nucleotides are chemically modified to increase the stability, binding affinity, and specificity of the oligos (Bennett et al, 2021; Roberts et al, 2020). There are two modes of ASO action. Firstly, the "gapmer" ASOs, which contain a central DNA-based unmodified sequence flanked by two wings of modified nucleotides, recruit RNase H1 for the degradation of target RNAs. Secondly, steric-blocking ASOs, which are modified at all nucleotides, bind to the target RNA, affect RNA structures or block interactions with RBPs, but do not induce RNase H cleavage (Fig. 5) (Nikom and Zheng, 2023; Roberts et al, 2020).

Multiple degradative ASOs have been approved for clinical usage. For example, Tofersen is an ASO targeting mutant *SOD1* RNA for degradation to stop the production of toxic SOD1 proteins (Miller et al, 2022). Inotersen is approved for the treatment of familial amyloid polyneuropathy and cardiomyopathy via targeting transthyretin-encoding transcript for degradation (Benson et al, 2018). However, several highly anticipated ASO drugs, namely one targeting *Htt* in Huntington disease (Tominersen) and two targeting the *C9ORF72* sense repeat-containing transcript in C9ORF72-ALS/FTD (IONIS-C9Rx/BIIB078 and Wave WVE-004), all failed in clinical trials (Kingwell, 2021; Kwon, 2021; van den Berg et al, 2024), despite success in preclinical studies with mouse models (Jiang et al, 2016; Kordasiewicz et al, 2012; Liu et al, 2022; Tran et al, 2022). There could be multiple reasons that need to be explored, including toxicity contribution from the antisense repeat and haploinsufficiency, among others. In addition, the clinical trial of the ASO targeting *ATXN2* (i.e., ION541/BIIB105) (Scoles et al, 2017) was discontinued due to its limited rescue effect on neurofilament and clinical measures. Promisingly, many other ASOs are being subjected to clinical trials with encouraging evidence. For example, IONISMAPTRx/BIIB080 targeting *MAPT* for AD treatment is at the phase-2 trial stage across multiple countries. ION859/BIIB094 targeting *LRRK2* for PD treatment is at the phase-1 trial stage, and the results are expected by the end of 2024. Moreover, ASOs targeting α-Synuclein showed beneficial effects in a rodent model of PD (Cole et al, 2021).

Steric-blocking ASOs can be used for diverse purposes (Fig. 5). The most widely used application of steric-blocking ASOs is splicing modulation. The first FDA-approved ASO drug for neurological diseases is Nusinersen (Spinraza), designed to treat SMA (Finkel et al, 2017; Neil and Bisaccia, 2019). The ASO is designed to bind and increase the exon 7 inclusion of *SMN2*, which can produce elevated levels of functional SMN protein to compensate for the loss of *SMN1*, the cause of SMA (Hua et al, 2010; Hua et al, 2007; Rigo et al, 2012). Thus far, there are a number of splicing-switching ASOs approved by the FDA for different neurological diseases, including four drugs for Duchenne muscular dystrophy (i.e., eteplirsen, golodirsen, casimersen and viltolarsen) (Baker, 2017; Dhillon, 2020; Heo, 2020; Pascual-Morena et al, 2020; Shirley, 2021), and a patient-customized

precision drug for the rare genetic neurodegenerative disease Batten disease (Kim et al, 2019). In addition, several steric-blocking ASOs are actively under development for various neurodegenerative diseases and have shown promising initial results. For example, ASOs were designed to inhibit cryptic exon inclusion in *STMN2* induced by loss of TDP-43, and they have been shown to restore the STMN2 protein level in mice (Baughn et al, 2023). *UNC13A* is another well-characterized TDP-43-linked cryptic exon-containing transcript. Using ASOs to inhibit the inclusion of the cryptic exon can rescue the UNC13A protein level and restore normal synaptic function (Keuss et al, 2024). For AD, ASOs that are designed to inhibit *MAPT* exon 10 inclusion have successfully achieved splicing alteration from toxic 4R to 3R in human tau-expressing mice (Schoch et al, 2016), suggesting important therapeutic potential.

Steric-blocking ASOs can also be designed for translation modulation. It is known that the 5'-UTR of an mRNA contains *cis*-acting elements that can modulate translation initiation (Chu and von der Haar, 2012). ASOs targeting the regulatory elements of the 5'-UTR showed selective effects on modulating protein translation (Hedaya et al, 2023; Liang et al, 2017b). However, no translation-modulating ASOs have been tested in neurodegenerative diseases so far. Furthermore, steric-blocking ASOs have been proposed to target aberrant miRNAs and prevent them from targeting mRNAs for decay (Lima et al, 2018).

RNA-targeting therapies exponentially broaden the spectrum of druggable targets by modulating the expression of coding and even noncoding genes. Given the wide recognition of RNA dysregulation in neurodegenerative diseases and the clinical success in some cases, RNA-targeting therapies hold great promise as candidates for precision medicine. However, several challenges remain in the clinical translation of RNA-targeting therapies (Box 1), such as the lack of efficient delivery methods to the central nervous system, undesirable immunogenicity, potential off-target effects, and uncertainties of long-term treatment outcomes. The continuous optimization and development of novel RNA-targeting approaches, including ASOs, siRNAs, small-molecule drugs, as well as viral and non-viral drug delivery methods, will provide safer and more effective options for modulating modifier genes. Beyond the advances in RNA-targeting and delivery methods, identifying the correct gene or pathway to target is particularly challenging for sporadic cases, which comprise most patients. Addressing this requires a deeper and more comprehensive understanding of the basic biology and fundamental molecular mechanisms underlying neurodegenerative phenotypes. The emergence of artificial intelligence (AI) is driving advances in drug design. A number of AI-identified candidate targets have been validated in ALS animal models (Pun et al, 2022; Zhang et al, 2022) and AD models (Merchant et al, 2023). Integrating interdisciplinary novel technologies that can address challenges beyond the reach of traditional methods, developing model systems that more accurately mimic human disease progression, utilizing large databases to predict critical pathogenic pathways or differentiate sub-groups of sporadic diseases, and exploring the genome and understudied candidate genes could provide valuable insights into modifying genes or pathways that may serve as potential therapeutic targets. Overall, there is great anticipation surrounding therapeutic approaches targeting RNAs of the candidate genes in treating neurodegenerative diseases.

# Peer review information

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

## Acknowledgements

This work is supported by NIH grants RF1NS113820 (SS), R01AG078948 (SS), RF1NS127925 (SS), and DOD grant HT94252410142 (SS). YL was a recipient of the Springboard Fellowship from the Target ALS Association. We thank the members of Sun lab for helpful discussions, with special thanks to Julie Asbury for proofreading. Figures were created with BioRender. Due to space limitations and the vast amount of research on this topic, some relevant references may be missing from this review.

## Author contributions

**Yini Li**: Writing—original draft. **Shuying Sun**: Supervision; Writing—review and editing.

## Disclosure and competing interests statement

The authors declare no competing interests.

