## [Peer Review File · The EMBO Journal]

RNA dysregulation in neurodegenerative diseases

Yini Li and Shuying Sun

Corresponding author(s): Shuying Sun (shuying.sun@jhmi.edu)

Review Timeline:

Submission Date:	19th Jun 24
Editorial Decision:	23rd Jul 24
Revision Received:	8th Oct 24
Editorial Decision:	12th Nov 24
Revision Received:	27th Nov 24
Accepted:	10th Dec 24

Editors: Kelly M Anderson and Ioannis Papaioannou

Transaction Report:

Dear Dr. Sun,

Thank you for submitting your review for consideration by the EMBO Journal. It has now been seen by one referee whose comments are enclosed. Given the referee's positive recommendations, I would like to invite you to submit a revised version of the manuscript, addressing the comments. Typically our reviews contain between 3-6 figures, so we ask that you please add two more figures where you see fit. These can be somewhat draft versions as graphics support is included in all review articles.

We generally allow three months as standard revision time. As a matter of policy, competing manuscripts published during this period will not negatively impact on our assessment of the conceptual advance presented by your study. However, we request that you contact the editor as soon as possible upon publication of any related work, to discuss how to proceed.

Thank you for the opportunity to consider your work for publication. I look forward to your revision.

Yours sincerely,

Kelly M Anderson, PhD
Editor, The EMBO Journal
k.anderson@embojournal.org

We realize that it is difficult to revise to a specific deadline. In the interest of protecting the conceptual advance provided by the work, we recommend a revision within 3 months (21st Oct 2024). Please discuss the revision progress ahead of this time with the editor if you require more time to complete the revisions. Use the link below to submit your revision:

Referee #1:

The review by Yini Li and Shuying Sun provides a comprehensive overview of the mechanisms and implications of RNA dysregulation in neurodegenerative diseases. The article is well-structured, informative, and contributes significantly to the field by discussing both established and emerging mechanisms. It covers a wide range of diseases, including Alzheimer's, Parkinson's, ALS, and Huntington's, highlighting shared and unique RNA-related mechanisms. The article effectively integrates recent findings with established knowledge, providing a thorough update on the field. It discusses potential RNA-targeting therapies, offering a hopeful perspective on future treatment strategies.

However, I have several major and minor comments and suggestions:

1. While the manuscript is comprehensive, it could benefit from deeper analysis. For example, exploring how the interplay between various steps of RNA metabolism contributes to disease phenotypes is missing. Additionally, throughout the manuscript, there is a strong focus on ALS and TDP-43. Perhaps this should be reflected in the title or at least in the abstract.
2. The visual aid accompanying the manuscript is below par. The figure itself only introduces various examples of dysregulated RNA metabolism but does not link them to corresponding diseases. It would be better to present a general scheme of RNA dysregulation's contribution to neurodegenerative diseases and provide more detailed figures representing various pathological mechanisms.
3. The review should expand on future research directions and the potential for clinical translation of RNA-targeting therapies.
4. In the first sentence of the abstract, it should be "years" not "year."
5. The introduction lacks references; this should be changed. All mentioned diseases and molecular phenomena should be appropriately cited.
6. Many cited studies are more than five years old. Thus, the statement in the introduction is misleading and should be changed.
7. On page 7, line 140, there is mention of an unpublished report. Is it submitted as a preprint? If yes, it should be acknowledged. If not, it should be removed from the review or the sentence should be more tentative.
8. On page 10, in the paragraph about NMD, incorrect publications are cited. The original publications by Maquat and Jacobsen should be cited.
9. On page 14, line 303, instead of the expression "in," it should be "and."
10. miRNAs should be introduced before circRNAs, as the biological role of circRNAs is often connected with miRNAs.
11. On page 22, line 489, in the sentence "We recently found...," a citation is missing.
12. The title of the section should be "Biomarkers and Therapeutics."

Reviewers' Comments:

We would like to thank the reviewer for the very constructive comments and suggestions, which help us significantly improve this review manuscript. We now have edited the writing as suggested. We also added two more figures. The editing was marked with yellow highlight.

Referee #1:

The review by Yini Li and Shuying Sun provides a comprehensive overview of the mechanisms and implication of RNA dysregulation in neurodegenerative diseases. The article is well-structured, informative, and contributes significantly to the field by discussing both established and emerging mechanisms. It covers a wide range of diseases, including Alzheimer's, Parkinson's, ALS, and Huntington's, highlighting shared and unique RNA-related mechanisms. The article effectively integrates recent findings with established knowledge, providing a thorough update on the field. It discusses potential RNA-targeting therapies, offering a hopeful perspective on future treatment strategies.

Major remarks

(1) While the manuscript is comprehensive, it could benefit from deeper analysis. For example, exploring how the interplay between various steps of RNA metabolism contributes to disease phenotypes is missing. Additionally, throughout the manuscript, there is a strong focus on ALS and TDP-43. Perhaps this should be reflected in the title or at least in the abstract.

Response: We thank the reviewer for the suggestion. We improved the discussion of interplay between various steps of RNA processing in each section and in figures. We agree there is more information about TDP-43 and ALS. We now mentioned in the overview. One reason is that there is a lot progress in understanding RNA metabolism related to TDP-43. There are relatively more studies directly linked to RNA metabolism in ALS compared to other neurodegenerative diseases. We did not include studies only using RNA-seq to identify gene expression profile changes in diseases if no specific RNA processing pathway was explored.

(2) The visual aid accompanying the manuscript is below par. The figure itself only introduces various examples of dysregulated RNA metabolism but does not link them to corresponding diseases. It would be better to present a general scheme of RNA dysregulation's contribution to neurodegenerative diseases and provide more detailed figures representing various pathological mechanisms.

Response: Thank you for this important point. We have now included three figures demonstrating the dysregulated RNA processing pathways, potential causes of RBP dysfunction and RNA-targeting therapeutics. We did not separate by diseases because RBPs are often found to be multi-functional, multiple RNA processing pathways could be dysregulated in one disease, and pathology of one RBP (such as TDP-43) could be found in multiple diseases. Therefore, we think it is clearer to organize the figure by pathways.

(3) The review should expand on future research directions and the potential for clinical translation of RNA-targeting therapies.

Response: Thank you for this suggestion. We now discussed future research directions at the end of each section. We also significantly expanded the discussion on RNA-targeting therapies.

Minor remarks

(1). In the first sentence of the abstract, it should be "years" not "year."

Response: Thank you. The typo has been corrected.

5. The introduction lacks references; this should be changed. All mentioned diseases and molecular phenomena should be appropriately cited.

Response: Thank you for pointing this out. We apologize for our previous oversight in failing to include citations for the introduction section. We now have added the references.

6. Many cited studies are more than five years old. Thus, the statement in the introduction is misleading and should be changed.

Response: We agree the statement is not accurate. We have changed to “We review recent discoveries alongside previous key findings”.

7. On page 7, line 140, there is mention of an unpublished report. Is it submitted as a preprint? If yes, it should be acknowledged. If not, it should be removed from the review or the sentence should be more tentative.

Response: The work was just published, so the articles are now cited in the review. We thank the reviewer for pointing this out.

8. On page 10, in the paragraph about NMD, incorrect publications are cited. The original publications by Maquat and Jacobsen should be cited.

Response: Thank you. We have corrected the citation and included those foundational works.

9. On page 14, line 303, instead of the expression "in," it should be "and."

Response: We apologize but we didn't find this at line 303.

10. miRNAs should be introduced before circRNAs, as the biological role of circRNAs is often connected with miRNAs.

Response: Thank you for this valuable suggestion. We have re-ordered the two parts.

11. On page 22, line 489, in the sentence "We recently found...," a citation is missing.

Response: Thank you for pointing it out. We have now added the missing citation.

12. The title of the section should be "Biomarkers and Therapeutics."

Response: Thank you for the suggestion. We agree that the section title "Biomarkers and Therapeutics" will be more appropriate, so we have changed the title as suggested.

Dear Dr. Sun,

Thank you again for submitting your revised manuscript to The EMBO Journal for our consideration, and for your patience. I am glad to say that all referee's concerns have been satisfactorily addressed in your revised manuscript, and we will therefore be happy to proceed with acceptance of your Review article for publication in The EMBO Journal as soon as the following formatting/editorial requests have been addressed in a final version of your manuscript:

1. Our team has carefully read your revised manuscript and corrected numerous grammar or other errors and typos throughout the text, made several suggestions for minor textual improvements, and asked for clarification/rephrasing in some cases (please find all edits and comments in the attached version of the manuscript file). Please note, however, that more corrections might still be needed. We kindly request you to carefully read through our attached file, accept all edits you agree with, and improve clarity where requested. Please also proofread your manuscript again and correct any remaining mistakes. We would also strongly encourage you to have your manuscript double-checked by native speakers of English, as we think that could improve the style of the text and increase the overall impact of your Review article.

2. Please provide a list of up to 5 relevant keywords after the Abstract of your revised Review article.

3. Please add a text box (named "Box 1" and called out at least once in the main text, where appropriate) with a summary - preferably in the form of bullet points- of particularly relevant open questions and future directions in the field that you think should be prioritized. You could also add your own suggestions in this text box of what could/should be done to address each one of the listed questions. Please name this Box "Questions and future directions" or "In need of answers" and place it at the end of the main text but before the list of References (for a couple of published examples, please refer to <https://www.embopress.org/doi/full/10.1038/s44318-024-00057-w> and <https://www.embopress.org/doi/full/10.1038/s44318-024-00109-1>).

4. You are welcome to include an Acknowledgements section, before the list of References, including also your funding information. Please note that the same funding information should be entered in our online manuscript tracking system during re-submission of your manuscript.

5. Please also include a conflict-of-interest statement, following your Acknowledgements section. The heading of this statement should be "Disclosure and competing interests statement".

6. Your Figure 1 contains too much information for a single Figure, and the dimensions are beyond our maximum Figure dimensions (i.e., it is too wide). Please consider moving some of its panels to a new Figure (or even two new Figures, if you prefer), and update the Figure callouts throughout the main text accordingly.

7. Please make sure that all Figures are as accurate as possible, and that all of their graphical elements are clearly defined/explained in detail in the respective legends.

8. Please also note regarding Figures:

- If there are certain aspects of your Figures that are based upon assumptions or where the scientific data remain ambiguous (for example, schematically depicting a presumed direct protein-protein interaction, protein shape or subcellular localization etc.), please add a comment so that we can work with you on an accurate depiction. Please ensure that the directionality and nature of interactions is presented accurately.

- If the figure or single panels of the Figure have been adapted from a published Figure, please add this information to the Figure legend (e.g., 'Adapted from...' or 'Based on...'). The editor will then discuss with you if a reference and permission will be necessary.

- Please only re-use Figures or parts of a Figure if this is essential for understanding the concept communicated. Often a reference to a previous paper will suffice. If the Figure contains re-used images or elements of images, including schematics, micrographs or photos, please make sure that you have the permission/license to publish them (this also applies to your own previous work, if the journal you previously published in retains copyright). Certain "creative commons" open access licenses, such as CC-BY 4.0, allow re-use without additional formal permissions. All re-used material must be explicitly cited.

- If you use an image database for scientific iconography (such as BioRender), please let us know if you have a license that allows for publication in an academic journal. If this is the case, the database should be acknowledged in the respective Figure legends.

- Please ensure that the information shown is scientifically accurate.

Please also note that as part of the EMBO publications' Transparent Editorial Process, The EMBO Journal publishes online a Peer Review File along with each accepted manuscript. This File will be published in conjunction with your Review article and will include the referee report, your point-by-point response, and all pertinent correspondence relating to the manuscript. You can opt out of this by letting the editorial office know (contact@embojournal.org). If you do opt out, the Peer Review File link will point to the following statement: "No Peer Review File is available with this article, as the authors have chosen not to make the review process public in this case."

We look forward to a minor revision of your Review article addressing the above points as soon as possible. Please let us know if you have any questions or comments you would like to discuss with us. When you are ready to re-submit your revision, please use the link:

<https://emboj.msubmit.net/cgi-bin/main.plex>.

Best regards,

Ioannis

All editorial and formatting issues were resolved by the authors.

Dear Dr. Sun,

I am very pleased to inform you that your Review article has been accepted for publication in The EMBO Journal. Thank you for addressing the referee concerns and the editorial and formatting requests.

Your manuscript will be processed for publication by EMBO Press. It will be copy edited and you will receive page proofs prior to publication.

I would also like to mention that we have sent the figures of your Review article to a professional scientific illustrator, who has already sent you the improved figures for your approval. Please send any requests for corrections you might have to the illustrator as soon as possible, so that we can move forward with the preparation of your manuscript for publication.

Important: Please note that you will be contacted by Springer Nature Author Services to complete licensing and payment information. Since this is a commissioned Review article, EMBO Press will cover all publication fees.

If you have any questions, please do not hesitate to contact the Editorial Office. Thank you for your contribution to The EMBO Journal!

Best regards,

Ioannis
